# On Mode Water formation and erosion in the Arabian Sea: Forcing mechanisms, regionality, and seasonality

Estel Font[1], Sebastiaan Swart[1,2], Puthenveettil Narayana Vinayachandran[3], & Bastien Y. Queste[1]

[1]Department of Marine Sciences, University of Gothenburg, Gothenburg, Sweden
[2]Department of Oceanography, University of Cape Town, Rondebosch, South Africa
[3]Centre for Atmospheric and Oceanic Sciences, Indian Institute of Science, Bengaluru, India

*Correspondence to*: Estel Font (estel.font.felez@gu.se)

**Abstract.** Mode water acts as a barrier layer controlling surface-to-interior fluxes of key climatic properties. In the Arabian Sea, mode water stores heat and provides an oxygen-rich layer for rapid remineralization, and its subduction is a direct pathway for oxygen into the upper Oxygen Minimum Zone. We use float observations to characterize the properties of the Arabian Sea mode water layer (MWL). The MWL forms when springtime warming stratifies the surface layer and caps the deep surface mixed layer formed during the winter monsoon. During the summer monsoon, a second MWL is formed south of 20ºN following the cessation of wind-driven mixing. We use a 1D and 3D model to disentangle the contributions from atmospheric and oceanic forcing on this water mass. The 1D model accurately represents the mode water's formation and erosion, showing that atmospheric forcing is the first-order driver, in agreement with observations. However, there are regions where advective processes, eddy mixing or biological heating are essential for the formation and/or erosion of the MWL. For instance, in the eastern Arabian Sea, freshwater-driven stratification advected via the West Indian Coastal Current reduces the potential for deep mixed layers via convective mixing, resulting in a thinner MWL. The 3D model shows that the MW contributes 5±1% to the oxygen content of the upper ocean, with its maximum during spring in the northern Arabian Sea (40±17%), thus highlighting the key role of the water mass in storing and transporting heat and oxygen to the interior.

## 1 Introduction

Mode waters (MWs) are a vertically homogeneous water mass formed in the surface mixed layer and when subducted, play a critical role in the ocean as a stratification barrier between the surface and ocean interior (Hanawa & Talley, 2001). Surface water properties (e.g heat, oxygen, organic matter) are physically transported and subsequently isolated in deeper layers by alternating mixing and stratification periods, trapping MWs (Lacour et al., 2023; Z. Li et al., 2023; Portela et al., 2020), possibly long-term (i.e. in favor of heat uptake, carbon sequestration, and oxygen ventilation). Despite occupying only 20% of the upper 2000 m of the ocean, mode water shows the most remarkable oceanic heat content change over the Argo era (2005–2020, Z. Li et al., 2023). Oxygen concentration below the surface is largely determined by the formation of MW and its subduction into the ocean interior (Hanawa & Talley, 2001; Portela et al., 2020). Moreover, MWs have a major influence on nutrient distribution as they prevent the upwelling of deep-sea nutrients into the photic zone (Bushinsky & Cerovečki, 2023) and are a key component of the mixed layer biological pump, an important pathway in the global carbon cycle (Dall'Olmo et al., 2016).

However, climate change is expected to alter MW formation. Predicted shoaling of surface mixed layers and intensified ocean stratification are projected to reduce MW formation (Gao et al., 2023; Xu et al., 2013). Thus, characterizing MWs in regions where stratification is strongly influenced by climate forcing (Albert et al., 2023; Mohan et al., 2021; Nisha et al., 2024) is essential for understanding their current impacts and how they may evolve in the future. Given their role in ocean heat uptake, biogeochemical cycling, and carbon sequestration, the life cycle of MW and its changes have significant implications for both regional and global ocean-climate dynamics.

Upper ocean properties and stratification have been widely studied in the Arabian Sea, (e.g. Babu et al., 2004; Lee et al., 2000; Prasad, 2004; Rao et al., 1989; Singh et al., 2019); yet no extensive literature exists on MW characterization in the region. McCreary et al. (1993) proposed the existence of a "fossil layer" which controls entrainment and detrainment allowing the exchange of mass, momentum and heat between the surface and the ocean interior due to winds or surface buoyancy fluxes. Hanawa & Talley (2001) defined a weak MW type in a high evaporation region, where high-salinity surface waters and subducted saline subsurface layers occur. Literature about this water is centered on the characteristics and formation of Arabian Sea High-Salinity Water (ASHSW). ASHSW is formed in the northern Arabian Sea at depths of 0–150 m after convective mixing during winter (Kumar & Prasad, 1999; Morrison, 1997; Prasad & Ikeda, 2002a, 2002b; Shenoi et al., 1993). It then advects equatorward along the 24 isopycnal surface as a subsurface salinity maximum (Han & McCreary, 2001; Prasad & Ikeda, 2002a). Zhou et al. (2023) show that in the northern Arabian Sea, the subduction of oxygenated ASHSW supplies oxygen to the OMZ and controls the seasonal depth of the upper OMZ oxycline. Liu et al. (2018) assessed regional subduction and obduction ventilation patterns in the North Indian Ocean using gridded reanalysis products, revealing that ventilation can occur during both winter and summer in a monsoon-dominated ocean. This fact suggests that the formation of MWs does not only occur in the north and is not only constrained to winter as ASHSW descriptions suggest. Hence, a comprehensive understanding of (a) MW formation and erosion especially their regional patterns and quantitative contributions from buoyancy forcing, winds, and advective processes, (b) MW properties and its fate, and (c) its interannual variability in the Arabian Sea are still lacking.

The Arabian Sea hosts the most intense oxygen minimum zone (OMZ) in the world, with suboxia prevailing across most of the intermediate ocean (150–1,250 m). It is characterized by a tight balance between biological and physical oxygen supply and consumption mechanisms, modulated by the monsoon cycle (Acharya & Panigrahi, 2016; McCreary et al., 2013; Rixen et al., 2020). Upper ocean stratification (and MW as one of its components) is a key factor in understanding heat (Nisha et al., 2024), oxygen (Ditkovsky et al., 2023; Lachkar et al., 2021), and net primary productivity (Wiggert et al., 2005) in the Arabian Sea. For instance, MWs have been shown to serve a unique role above oxygen minimum zones whereby they act as a reservoir of oxygen below the productive layer (Kalvelage et al., 2015) and thus modulate the upper boundary of the OMZ. Moreover, carbon exported from the surface transits through the MW where the most labile organic matter is rapidly remineralized (Weber & Bianchi, 2020). The thickness, properties, and fate of this MW govern how much carbon is remineralized therein, how

recalcitrant is the organic matter exported to the OMZ, and whether the respired $CO_2$ is returned to the surface with the deepening of the surface layer, or subducted into the OMZ for long term sequestration (Weber & Bianchi, 2020). MWs are also key to the heat budget of the upper ocean, as a barrier between the warm surface and cold interior. MW presence can potentially maintain the Arabian Sea Warm Pool by effectively inhibiting the vertical mixing of the upper ocean with deeper cooler waters (similar to barrier layers formed by freshwater advection from the Bay of Bengal in the southeast Arabian Sea; Li et al., 2023).

There is a need for understanding the fate of MW, which will determine if subducted properties from the surface will either be returned to the surface or sequestered down into the mesopelagic with long-term and poorly understood impacts on regional oxygen, carbon, and heat budgets. In this study, we use observational Argo data to characterize the properties and timing of MW formation and erosion. We employ a one-dimensional model (GOTM) to assess the main forcing components involved in the MW's life cycle, and a 3D regional model to upscale these findings to assess MW contribution to the oxygen budget of the Arabian Sea. This approach provides a more comprehensive understanding of the processes governing MW dynamics in the Arabian Sea.

## 2 Data

### 2.1 Argo

We use the array of Argo profiling floats to characterize MW in the Arabian Sea. Between 2000 and 2023, a total of 579 floats sampled in our area of study (0-30ºN, 30-80 ºE, Figure 1a), providing 99355 profiles of temperature, salinity, and pressure that were flagged as "good" by the CORIOLIS Data Centre over the studied period (Gaillard et al., 2009). The data are binned every 4 m in depth up between 0-500m and at 10-day resolution. The coverage of Argo floats over the Arabian Sea extends over the whole domain, with an average of 30 profiles per 0.5ºx0.5º, except the shallower, near coastal regions of the Indian subcontinent (Figure 1a). The northern Arabian Sea has the most profiles, up to 130 profiles per 0.5ºx0.5º in the Sea of Oman (Figure 1a).

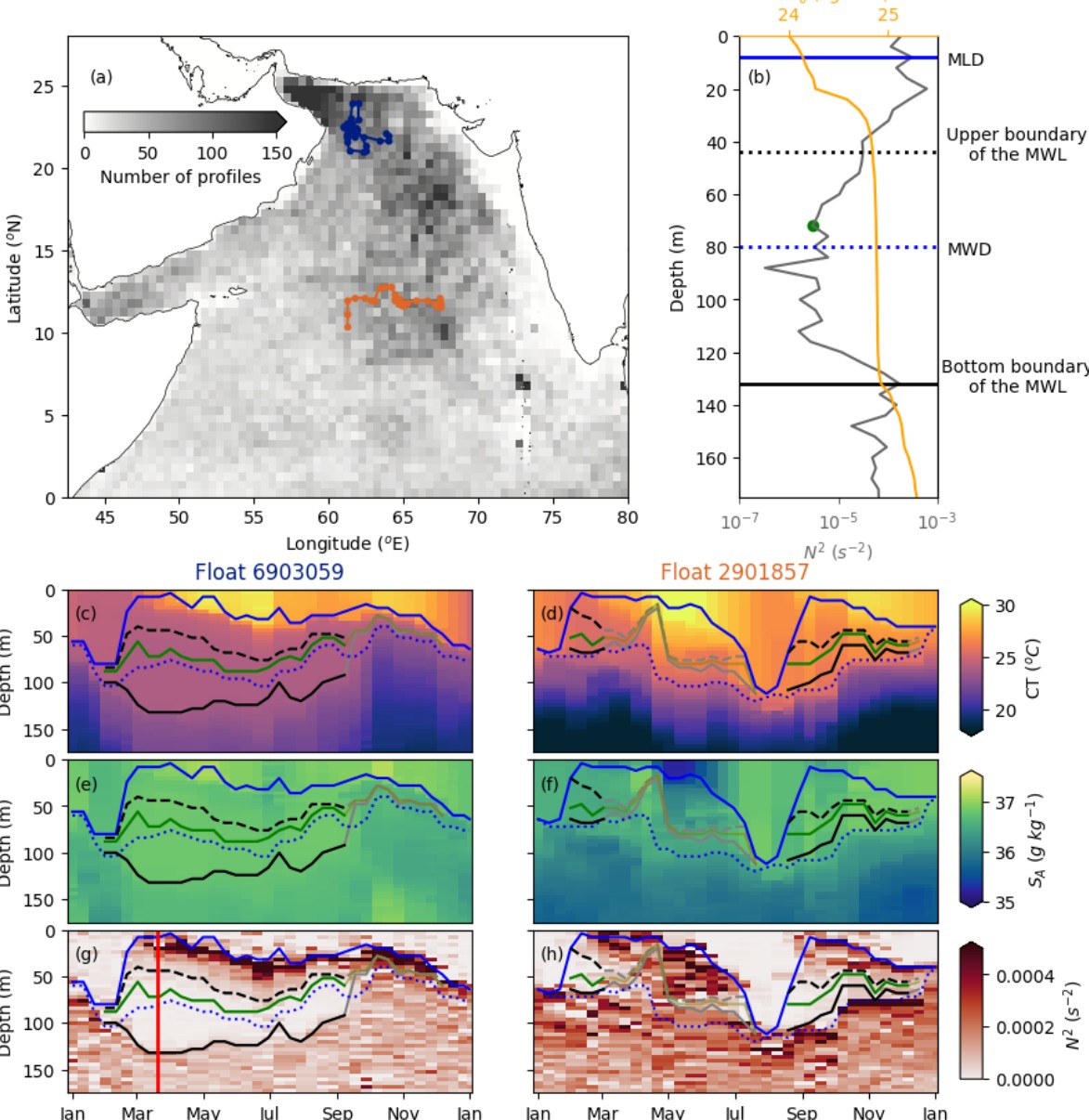

90

**Figure 1. Data and mode water definition. (a)** The total number of argo profiling floats in the Arabian Sea between 2007 and 2023 on a 0.5°x0.5° grid. The trajectory of float No. 6903059 during 2022 in the northern Arabian Sea is in blue and the trajectory of float No. 6903059 during 2018 in the southern Arabian Sea. **(b)** Example profile (marked in red in panel g) of stratification ($N^2$, grey) and potential density ($\sigma_0$, yellow). Mixed layer depth (MLD) is indicated with a blue line, mode water depth (MWD) in the dotted blue line, the core of the mode water layer is the green dot, and the upper and bottom boundary of the mode water layer are a dashed and solid black line, respectively. **(c-h)** Example of two floats annual cycle of conservative temperature (CT, c-d), absolute salinity ($S_A$, e-f), and stratification ($N^2$, g-h) in the northern Arabian Sea (No. 6903059, c-e-g) and southern Arabian Sea (No. 2901857, d-f-h). As per panel b, MLD is indicated with a blue line, MWD in the dotted blue line, the core of MWL in green, and the upper and bottom layer of the MWL as a dashed and solid black line, respectively. Grey lines indicate when MW is classified as eroded (see Section 3.1).

## 2.2 GOTM 1D model

The General Ocean Turbulence Model (GOTM; Burchard et al., 1999), one-dimensional water column model, is used to investigate the importance of upper ocean processes to the vertical stratification and MW formation and erosion in the Arabian Sea. The GOTM CVmix GCM mixing parameterizations (Li et al., 2021) are used. This is a proven stand-alone model for
studying boundary layer dynamics (e.g., Umlauf & Burchard, 2005). The models were initialized with the first profile of each float in the region and forced with 3-hourly heat, momentum, and freshwater fluxes from the ERA5 atmospheric reanalysis product (Hersbach et al., 2020), colocated to the time and location of the floats. The model setup has a vertical resolution of 2 m and 3-hourly time steps. Background interior mixing and shear mixing are parameterized in the CVmix GCM using the KPP mixing scheme (K-profile parameterization; Large et al., 1994); Langmuir turbulence is parameterized as per Li et al. (2016),
and we neglect double diffusion. The optical properties of different water types influence the absorption of shortwave radiation, potentially modifying stratification. Absorption coefficients vary due to chlorophyll-driven light attenuation, which is relevant across all regions but depends on local phytoplankton concentrations. Sediment-related absorption is primarily a concern in near-shelf environments and is unlikely to significantly influence offshore regions. This study uses the Jerlov II water type scheme (Qing et al., 2022; Sasmal, 1997), which provides the best representation of mixed layer depth (MLD) compared to
observational data and aligns with past observations of optical properties (Qing et al., 2022; Figure S1). However, this represents a limitation, as it may introduce biases in regions where optical properties change seasonally. The potential impact of these biases on vertical stratification is examined in our model comparisons.

## 2.3 MOM4p1 - TOPAZ 3D model

The physical model output used in this study is based on the Modular Ocean Model (MOM4p1; Griffies et al., 2003) while this specific configuration extends from -30 to 30ºN and 30 to 120ºE (Behara & Vinayachandran, 2016; Kurian & Vinayachandran, 2007). This configuration has a 1/4º spatial resolution and 40 vertical levels, with 25 levels in the top 200 m. The vertical grid has a 5 m resolution in the upper 60 m, 10 m resolution to 100 m depth, 20 m resolution to 200 m depth, and 700 m resolution to 5000 m depth (Kurian & Vinayachandran, 2007). A combination of Laplacian and biharmonic schemes
with Smagorinsky coefficients are used for horizontal mixing (Griffies & Hallberg, 2000) and the vertical mixing is based on the KPP (K-profile parameterization) scheme of Large et al. (1994). The physical model was started from a state of rest and spun up for ten years, and the coupled biophysical model for another 10 years using the climatological forcings. Further, the interannual run is forced by ERA-INTERIM three-hourly fields (Dee et al., 2011). The penetrative shortwave radiation parametrization is based on Morel & Antoine (1994), which depends only on the observed chlorophyll concentration (Sea-
Viewing Wide Field-of-View Sensor (SeaWiFS); Hooker & Esaias, 1993). The coupled ecosystem model is based on the

Tracers of Phytoplankton with Allometric Zooplankton (TOPAZ; Dunne et al., 2010). The coupled model provides a realistic representation of the seasonal cycle of temperature, salinity, currents, and oceanic productivity and its variability in the Indian Ocean (Kurian & Vinayachandran, 2007; Vijith et al., 2016). The model has been used successfully to study mixed layer processes (Kurian & Vinayachandran, 2006), the Arabian Sea warm pool (Kurian & Vinayachandran, 2007), freshwater plumes and advection from the Bay of Bengal (Vinayachandran et al., 2007), biophysical interactions associated with climate modes (Park et al., 2014) and phytoplankton bloom in the northern Arabian Sea (Vijith et al., 2016). Hence, the evaluation of model physics and biogeochemistry is not presented here. We use the quarter degree, 3-day resolution output from 2000-2020 (Prasanth et al., 2021).

## 3 Methods

### 3.1 Mode Water Layer Definition

The mode water layer (MWL) is the subsurface well-mixed layer created from capping of deep mixed layers by new near-surface stratification. We developed an algorithm to track this layer inspired by previous subtropical mode water studies (Feucher et al., 2019). We define the Mode Water Depth (MWD; dotted blue in Figure 1b-h) as the densest isopycnal of the surface mixed layer. The MLD is defined using a surface density threshold of 0.125 kg m$^{-3}$ and 2.5 m reference depth (solid blue in Figure 1b-h; Font et al., 2022; Liu et al., 2018; Montégut et al., 2004). The MWL is defined as the subsurface well-mixed layer capped after the deep seasonal surface mixed layer. The core of the MWL is defined as the minimum of N$^2$ between the MWD and the MLD (green in Figure 1b-h). We define the upper and the bottom boundary of the MWL using a mixed layer threshold method (0.05 kg m$^{-3}$) on either side of the core of the MWL in depth space: upwards to find the upper boundary (dotted black in Figure 1b-h), and downwards to find the bottom boundary of the MWL (solid black in Figure 1b-h). This is analogous to a typical surface boundary layer MLD detection (Montégut et al., 2004). While the algorithm does not explicitly track advection, the density threshold allows for the detection of low-stratified MWL that has formed elsewhere and been advected into the region. The resulting metrics are in deepening order: the surface, the MLD, the upper boundary of the MWL, the core of the MWL, the bottom boundary of the MWL, and the MWD (Figure 1b). Mode water thickness (MWT) is defined as the difference between the bottom and the upper boundary of the MWL (MWT=bot. MWL - up. MWL).

The formation time of the MWL is defined as the time of detection of the densest water during deep surface mixed layers that define the MWD. The erosion time of the MWL is defined as when the stratification at the core of the MWL is larger than $5x10^{-5}$ s$^{-2}$ (N$^2_{MWL}$ > $5x10^{-5}$ s$^{-2}$) or the thickness of the MWL is thinner than 10 m (MWT < 10 m) sustained for 30 days. The N$^2_{MWL}$ threshold represents the approximate background stratification in the region, ensuring that MWL erosion is defined relative to physically meaningful conditions. The MWT threshold ensures sufficient vertical data coverage given the 4 m depth resolution, while the 30-day duration is chosen to account for the 10-day temporal resolution and minimize the impact of data

gaps. Throughout this study, we use the definition of seasons based on three-month periods: winter monsoon (December–February, DJF), spring (March–May, MAM), summer monsoon (June–August, JJA), and fall (September–November, SON).

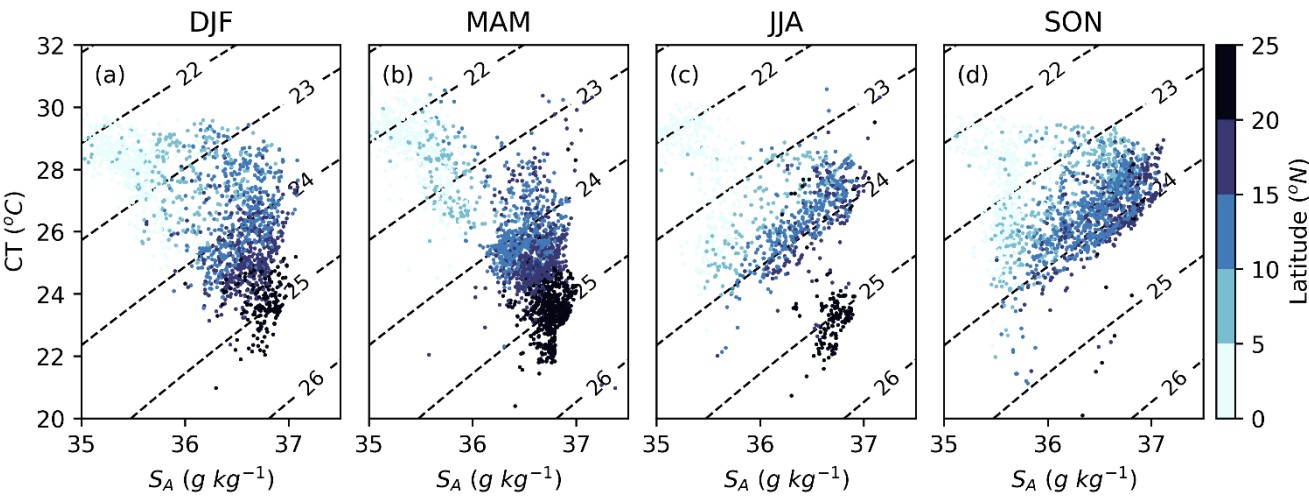

**Figure 2. Mode Water seasonal and spatial variability.** Mode water core T-S diagram from all available float profiles in the Arabian Sea for (a) winter (DJF, December-January-February), (b) spring (MAM, March-April-May), (c) summer (JJA, June-July-August), and (d) fall (SON, September-October-November) colored by latitude. Isopycnals (dashed black) are shown for all panels.

## 4 Results

### 4.1 Mode Water Cycle in the Arabian Sea: Seasonality and Regionality

Mode waters' life cycle in the Arabian Sea presents two different regimes dependent on latitude, linked to regional variability of the monsoons. This asymmetry results in MW formation occurring once a year in the north following deep winter mixed layers (Figure 1ceg, latitude < 20 °N), and twice a year in the south after both the winter and summer monsoons (Figure 1dfh, latitude > 20°N). In this section, we characterize the timing of formation, erosion, duration, and thickness for the contrasting monsoons and regions. Figure 2 provides an overview of the MW T-S properties colored by latitude, serving as the basis for contrasting the North, Central, and South Arabian Sea. Building on this, Figure 3 further illustrates the duration of the MWL, MWT, and the seasonality of the MW volume, while Figure 4 highlights MW formation relation to atmospheric forcing.

**In the northern half of the Arabian Sea** (latitude > 20°N), MW forms during the winter monsoon (Figure 1ceg, 2a, 3c-e, and 4a). The MWL is formed between January and March (Figure 3cd). The densities of the core of the MWL range between 24.5-

25 kg m$^{-3}$ (CT-S$_A$, 22-24 °C, 36.6-37 g kg$^{-1}$, Figure 2ab). It is present for over three months (94±28 days, Figure 3a) with an average maximum thickness of 41±15 m. In the Sea of Oman, MWL is thickest (> 50 m, Figure 3c-e) and present longest until it erodes on average in May-June (> 4 months, Figure 3a). MW has been observed up to September (Figure 1ceg). During the summer monsoon, the remnants of the winter MWL are still present and not fully dissipated. There is no MW formation during summer (Figure 2c and 3j-k) due to strong positive buoyancy gain (Figure 4b).

**In the central Arabian Sea** (10-20 °N), MW forms during both winter and summer monsoon. After the winter formation, the MWL is thinner in the east (Indian coast, 20±5 m) and west (Omani coast, 26±9 m) of the basin, and they dissipate in April (Figure 3f). In the center of the basin, the MWL is thicker (30±12 m on average, Figure 3d) and is present until April with some remnants in May (Figure 3g), dissipating from south to north. MW properties in the central Arabian Sea during winter formation are lighter than in the northern Arabian Sea, ranging within 23-24.5 kg m$^{-3}$ with T, S ranging 22-26 °C, 36-37 g kg$^1$ (Figure 2ab). During the summer monsoon, MW forms again in August (Figure 1dfh and 2cd) and is thickest in the central region (28 ±10 m, Figure 3j), lasting for 2 months until October (52±12 days, Figure 3b). These MWs are lighter than in winter, ranging within 23-24 kg m$^{-3}$ (Figure 2c).

**In the southern Arabian Sea** (latitude < 10 °N), the MWL is thin (21±7 m) and dissipates fast (~30 days) resulting in a patchy signal when approaching the equator (Figure 3). Equatorial seasonally reversing winds drive complex patterns of upwelling and downwelling (Phillips et al., 2021) that might suppress its formation, as per the summer monsoon upwelling in the eastern coast of Oman. Despite this fact, there is a striking difference between the west and the rest of the southern basin during summer (Figure 3j-l). We hypothesize it is due to the longitudinal gradient in wind strength combined with the high energetics of the western region during this season (Beal et al., 2013; Phillips et al., 2021). MWL is present south of 10 degrees close to the Somalian coast and the "Great Whirl". In this region, the MWL is 23±10 m on average and lasts until October (50±9 days, Figure 3j-l). The MW in the southern Arabian Sea is the freshest of the MWs, lighter than 24 kg m$^{-3}$ with a T-S range between 24-30 °C, < 36 g kg$^{-1}$ (Figure 2). Closer to the equator, the MWL does not form because surface mixed layers are shallow (Figure 6a-d) due to positive buoyancy flux and low winds (Figure 4b).

**The annual cycle** of the volume of MW presents a clear bimodality following the monsoons (Figure 3o). The difference in areas covered by the north and south can bias our view on the importance of the northern Arabian Sea to the total MW volume. Thus, we build an impact factor scaling the volume of MW by the contribution of the northern and southern Arabian Sea to the total Arabian Sea area (Figure 3p). The impact factor of MW is computed as $Impact\ Factor_{region} = MW\ volume_{region} \cdot \left(Area_{Arabian\ Sea}/Area_{region}\right)$, where the MW volume for each grid cell is computed as the MWT

multiplied by the cell area; the Arabian Sea area is defined by the ocean area shown in Figure 3a and the region areas are the northern or the southern Arabian Sea delimited by the 20°N parallel. The northern Arabian Sea is the 10% and southern Arabian Sea the 90% of the total area. The impact of the northern Arabian Sea on the total volume of MW is larger than the south in the first half of the year, despite its area being 10% of the total Arabian Sea (Figure 3o). MW formed in the south during the summer monsoon impact is smaller than the northern Arabian Sea due to their localized presence in the center of the Arabian Sea. The impact factor highlights the importance of the northern Arabian Sea in the total volume of MW in the Arabian Sea. In the following sections, we disentangle the forcing mechanisms for formation and erosion in the contrasting monsoons and regions.

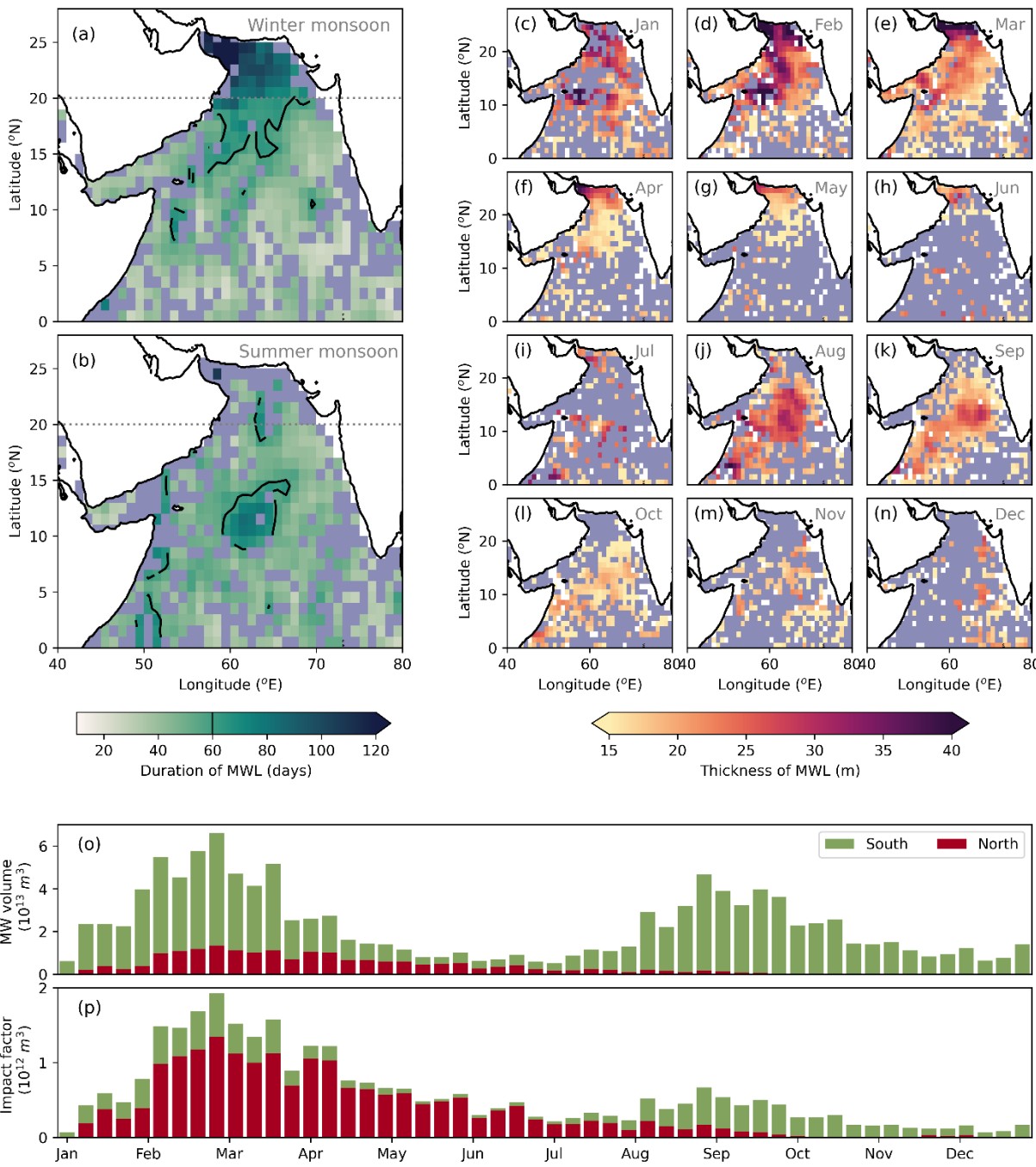

**Figure 3. Bimodal regionality of mode water formation.** (a-b) Mode water layer duration during the (a) winter monsoon and (b) summer monsoon. Black contours show regions where the duration is 60 days (2 months). (c-n) Monthly climatology of MW thickness where MW is present from float observations. Blue-shade is where there are float observations but MW is not detected. (o) Annual cycle of MW volume

in the Arabian Sea, colored by the contributions of the northern and southern Arabian Sea in red and green respectively. (p) The impact factor of MW is defined as MW volume scaled by the region's area contribution to the total Arabian Sea area.

## 4.2 Timing of Formation of MWL: Interplay of Surface Buoyancy Fluxes and Wind Forcing

Heat gain and freshwater input (rainfall) supply buoyancy to the ocean surface, stratifying the upper ocean, whereas heat loss
and evaporation result in buoyancy loss and convective mixing, deepening the surface mixed layer. Wind forcing also contributes to the mechanical mixing of the ocean boundary layer. The interplay between buoyancy fluxes (heat and freshwater, Equation 1) and winds determines the evolution of the surface mixed layer variability (Font et al., 2022; Niiler & Kraus, 1977; Singh et al., 2019) and we thus hypothesize that should also be the first-order drivers of the MWL life cycle in the Arabian Sea. In this section, we compare the climatological MW presence from float observations to the climatological atmospheric
data from ERA5.

The buoyancy flux through the surface (B) is used to determine the stability of the upper ocean and can be expressed as:

$$B = g \cdot \left[ \frac{\alpha \cdot Q_{NET}}{\rho_0 \cdot c_p} - \beta \cdot S_A \cdot (E - P) \right], \tag{1}$$

where g is the gravity constant, $\rho_0 = 1027$ kg m$^{-3}$ is the reference density, $c_p$ is the specific heat of seawater, $S_A$ is the median
absolute salinity between 10 and 15 m, $\alpha$ is the effective thermal expansion coefficient ($-\rho^{-1} \cdot (\partial \rho / \partial T)$), and $\beta$ is the effective haline contraction coefficient ($\rho^{-1} \cdot (\partial \rho / \partial S)$). $Q_{NET}$ has units W m$^{-2}$, E and P have units m s$^{-1}$, and B has units m$^2$ s$^{-3}$. In this representation, we do not include lateral processes such as horizontal advection, and mixing generated by horizontal processes.

We separate buoyancy from wind mixing terms to understand the timing of MW formation (Figure 4). The annual cycle of
MW presence, represented as the percentage of float profiles containing MW per latitudinal bands, shows the formation ones a year in the northern Arabian Sea and the biannual presence in the southern Arabian Sea (Figure 4a). We retrieve atmospheric data from ERA5 closest to the float profiles and construct the seasonality of buoyancy fluxes and wind stress over the same region (Figure 4bc). The buoyancy flux is driven by its thermal component ($\frac{\alpha \cdot Q_{NET}}{\rho_0 \cdot c_p} \gg \beta \cdot S_A \cdot (E - P)$, Figure S2). Deep surface mixed layers can be formed during both monsoons due to heat loss (Liu et al., 2018; Singh et al., 2019).


During the winter monsoon, MW formation is driven by surface buoyancy gain, which decreases surface density and stratifies the upper ocean (Figure 4), capping the residual mixed layer formed by convective mixing during winter. Low winds are present and those do not contribute to delaying the restratification after the change in the buoyancy flux regime (Figure 4c). The climatological average surprisingly shows the presence of MW before the buoyancy flux changes sign (Figure 4b). Despite
the apparent contradiction in the average climatologies, the contradiction disappears when considering the regional and

interannual variability on B and formation days (Figure 4d and S2). We present the histogram of the dates when B changes sign (light gray, Figure 4d) and the histogram of formation days (black line, Figure 4d). The interannual and longitudinal variability explains the spread of formation days that coincide with the spread of the days of change in the buoyancy flux sign (Figure 4d and S2).


During summer, in the northern Arabian Sea (latitude $> 20$ °N), there is no buoyancy loss and the ocean keeps gaining buoyancy (Figure 4b). Winds are stronger ($<0.05$ m s$^{-2}$) than the rest of the year, yet mechanical mixing cannot break the upper ocean stratification (Figure 4bc). On the contrary, in the southern Arabian Sea (latitude $< 20$ °N), strong monsoon winds from the southwest ($> 0.1$ m s$^{-2}$) and heat loss (-40 W m$^{-2}$) from the ocean to the atmosphere overcome the heating from solar radiation

and freshwater input from precipitation (Figure 4bc). In contrast to the formation of MW during winter, in summer the formation is shifted late after the change of sign of the buoyancy flux as the monsoon winds are still strong and mechanical mixing can maintain deep surface mixed layers (Figure 4d). When those winds die down, MW forms (Figure 4). Thus, MW does not form in the northern Arabian Sea during the summer monsoon, but is present in the southern Arabian Sea, resulting in the unique bimodal regionality.


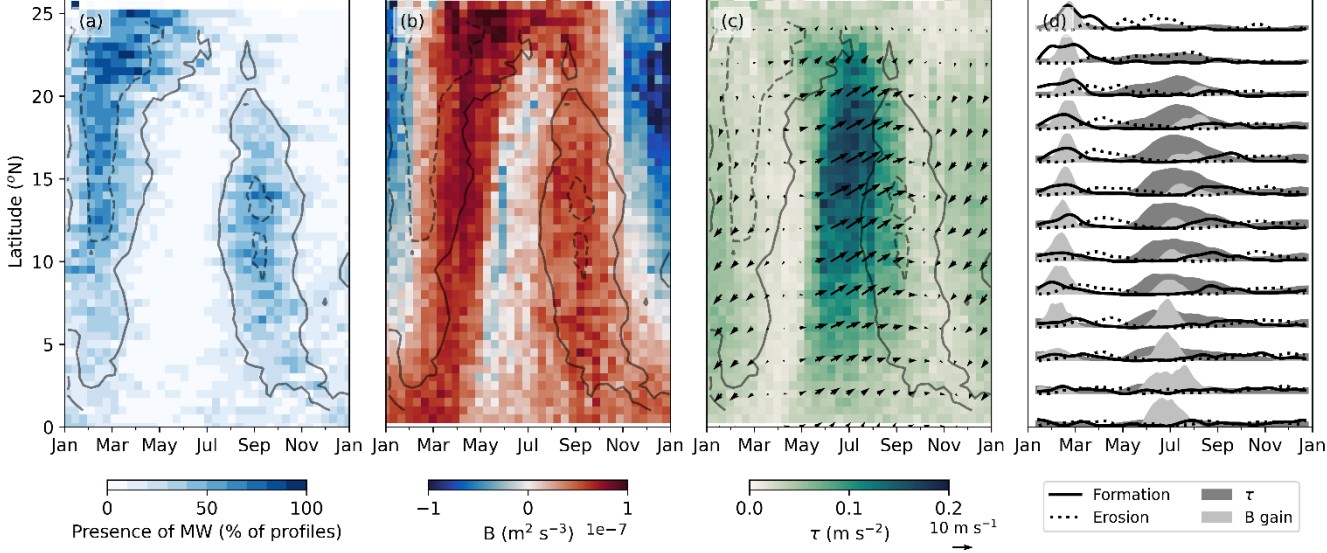

**Figure 4. Forcing mechanisms in the formation of MWL.** Latitudinal hövmoller of the climatology of (a) MW presence represented as the percentage of profiles containing MW, (b) surface buoyancy flux (B), (c) wind stress (τ) with arrows depicting wind speed and direction. Isoline of the percentage of profiles with MW presence is depicted as a gray solid (50%) and dotted contour (20%) in a, b and c. (d)

Distribution of the days of MW formation (black solid), days of MW erosion (black dotted) and days of change in buoyancy flux from loss to gain (shade light gray) per 2° latitudinal bands. Wind stress time series normalized by its maximum value per latitudinal band (shade dark gray).

**4.3 Erosion of the Mode Water Layer: Contrasting Monsoons**

The fate of MW determines if subducted properties from the surface will either be mixed back up as the surface mixed layer deepens or mixed down based on internal and poorly understood processes and instabilities (eg. internal waves, eddies, double-diffusion, fontal instabilities). In this section, we use observational data to evaluate the rate of change of the boundaries and regionality of the erosion rates of the MWL. Positive rates are shoaling of the boundary and negative rates are deepening of the boundary. Understanding how MW boundaries change over seasons is important for interpreting whether the capped

properties subduct or obduct.

Mode waters formed in winter and summer monsoons exhibit distinct patterns in how the MW boundaries change across seasons and regions (Figure 5). During spring, its upper boundary is on average 60±7 m, and the bottom boundary at 84±7 m (Figure 5ae) in the northern Arabian Sea, and a bit shallower for the southern Arabian Sea (upper boundary at 50±5 m and

bottom at 71±6 m). The MWL thins at an average rate of 15±4 m per season in the northern Arabian Sea and 10±4 in the southern Arabian Sea (Figure 5i). Both boundaries deepen (Figure 5cg), with the upper boundary at a faster rate, deepening more than 20 m per season (Figure 5c). This deepening is driven by the strengthening of spring stratification caused by warming from enhanced solar radiation at the ocean surface coinciding with low winds (Figure 4bc). Thus, during spring the deepening of the upper boundary contributes most significantly to the erosion rate of the MWL.


Following the summer monsoon's MW formation, the average rate of MW erosion is 6±4 m per season in the southern Arabian Sea (Figure 5j). During fall, MW boundaries are on average shallower than during spring: 45 ±10 m for the upper boundary and 65±11 m for the bottom boundary (Figure 5bf). In contrast to the boundary behavior following the winter monsoon, in fall both boundaries shoal (Figure 5dh), with the bottom boundary being the primary driver of the erosion of the MWL. It shoals

at an average rate of 20 m per season (Figure 5h). Although the erosion rates are similar or even faster during the winter monsoon compared to the summer monsoon (Figures 5ij), as the MWL is thicker during winter in the northern Arabian Sea (Figure 3 and 7a), this results in a longer MWL presence in this region (Figure 3b). Further investigation is needed to determine what are the mechanisms driving this erosion.

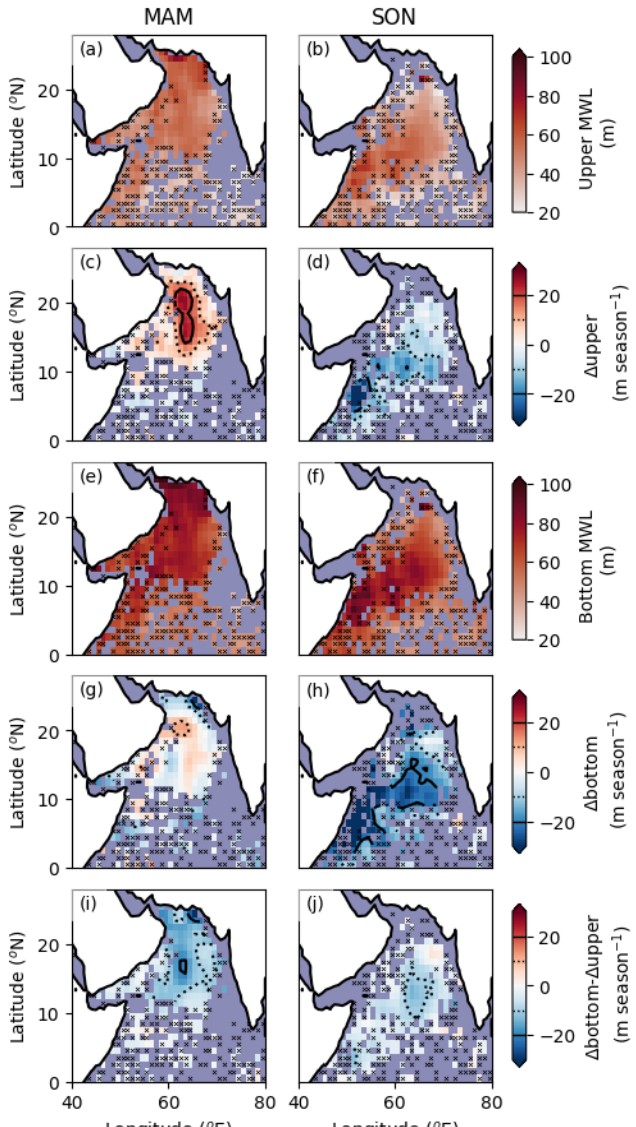


**Figure 5. Erosion of MWL.** Seasonal climatologies of the depth of the (a-b) upper boundary (Upper MWL) and (e-f) bottom boundary (Bottom MWL) for spring (a-e, MAM) and fall (b-f, SON). Depth change of the (c-d) upper boundary (Δupper) and (g-h) bottom boundary (Δbottom) of the MWL in m season$^{-1}$ for spring (c-g) and fall (d-h). Positive values mean deepening and negative is the shoaling of the boundary. (i-j) Rate of erosion per season (Δbottom - Δupper) for spring (i) and fall (j). Negative values mean the thinning of the MWL. The rates of change of ±20 m season$^{-1}$ (solid black) and ±10 m season$^{-1}$ (dotted black) are depicted. Crosses show the grid cells where less than three MWLs are detected.

## 4.4 Regionality: Role of vertical mixing and advective processes

Previous sections neglected horizontal processes assuming a dominance of top-down atmospheric forcing, showing that 1D processes are leading order in defining MW existence and lifespan. We now investigate potential contributions from 3-D physical processes (circulation, up-downwelling, shear) that can also alter upper ocean stratification. To investigate the MWL's formation and erosion mechanisms, we implement a 1D model (GOTM) to assess how well we can represent this layer by isolating vertical processes (1-D) from the three-dimensional (3-D) dynamics. From the seasonal difference maps between the 1D model and observations, we suggest mechanisms driving these differences for case-distinct regions in the Arabian Sea (e.g., misrepresentation of the mixed layer, stratification due to advected water masses, heat absorption due to biological presence, internal mixing), which are corroborated if such biases are absent in the 3D coupled model (MOM4p1 - TOPAZ) (Figure 6-7).

The surface mixed layer depth significantly influences the characteristics of the MWL. Therefore, we begin by analyzing how the model represents observed MLD patterns (Figure 6). Both 1D and 3D models show a bias towards deeper MLD compared to observations (Figure 6e-l). The largest bias is present during winter (DJF, 1D model 18±11 m; 3D model 20±9 m; Figure 6ei). The 1D model MLD bias displays regional variability, which could indicate that the 1D approach cannot fully represent surface mixed layer processes. In the 3D model, the shift of stratification to deeper than observed has been previously assessed by Prasanth et al. (2021) and in Supplementary Information Figures S3-S4. The MLD bias is consistent across the basin during winter, posing a constraint to our analysis. However, as the seasonal cycle is well reproduced, and the 3D model accurately captures the deep summer mixed layer in the center of the basin, we consider the model suitable for assessing MW life cycle mechanisms. For instance, horizontal advection, particularly in high eddy kinetic energy regions, redistributes heat and momentum laterally, a process absent in the 1D model. In the western Arabian Sea, where eddy kinetic energy intensifies in summer near the Somalian coast (Figure 1 in Zhan et al., 2020; Sun et al., 2022), this likely contributes to the MLD differences between the 1D and 3D simulations (Figure 6c,g,k).

Since the MLD determines the initial MWL thickness, biases in MLD directly affect the models' representation of MW thickness. For example, the 3D model overestimates the MLD during winter, leading to an overestimation of MWL thickness in spring (northern Arabian Sea: 24±5 m, southern Arabian Sea: 19±3 m; Figure 6ei and 7fj). Conversely, the 1D model underestimates the MLD, resulting in an underestimation of MW thickness, especially in the central basin during summer (Figure 6g and 7g). After summer deep surface mixed layers in the central Arabian Sea where MLD is underestimated (3±14 m, peaking to 24 m in the central Arabian Sea- blue dot; Figure 6cg), the 1D model also underestimates the thickness of the MWL (7±4 m, with 12 m bias in the central Arabian Sea - blue dot and case study in Figure 8). This underscores the critical importance of accurately modeling the MLD to represent MW dynamics reliably. The relationship between MLD and MW

biases warrants further investigation. As MLD is easier to diagnose in models, analyzing its regional biases and lagged relation to MWT biases could help refine model representation of this layer and better constrain MW ecological implications.

To better constrain the regional differences, we look at the vertical structure for selected case study regions (marked regions in Figure 6-7a). The vertical sections are climatological averages from all floats gridded 2° x 2° and the line plots are the depth

cumulative sum of the $N^2$ bias between observations and the 1D and 3D models (Figure 8). We identify four regions, which are (1) The Sea of Oman (SoO), (2) the Central Arabian Sea, (3) the Eastern Arabian Sea (EAS) and (4) the Southern Arabian Sea (SAS).

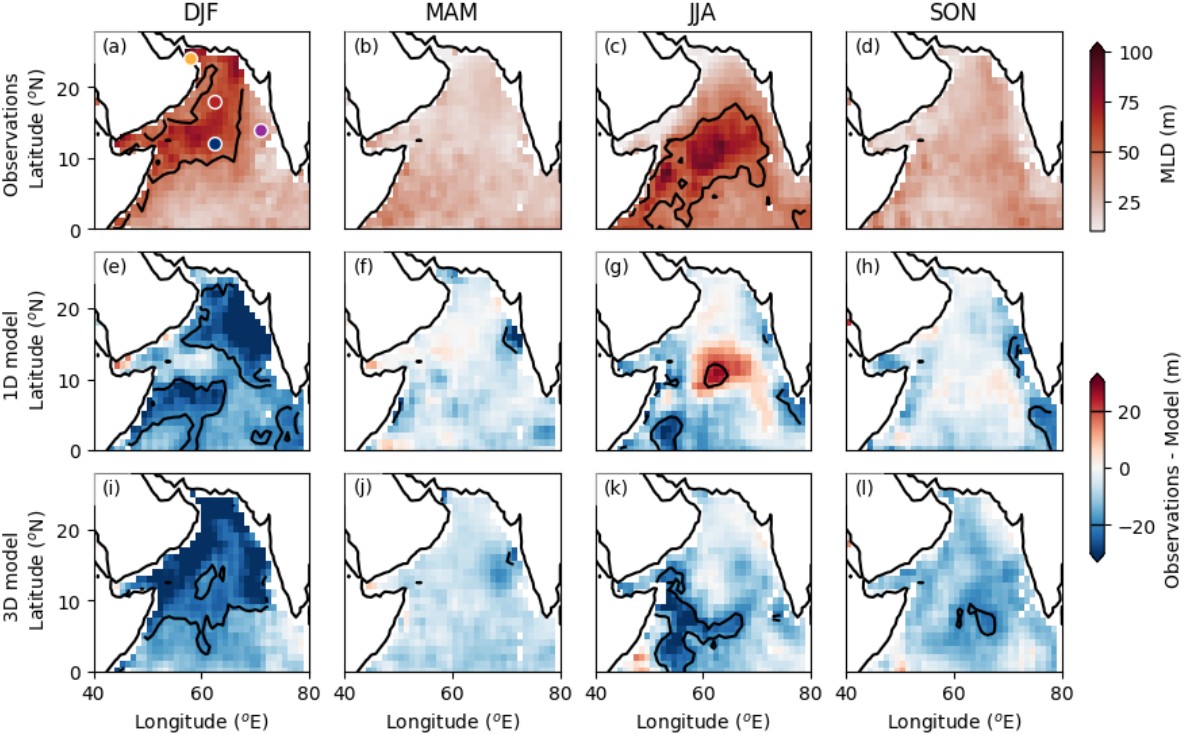


**Figure 6. MLD comparison between observations, 1D GOTM and 3D MOM4p1 - TOPAZ.** (a-d) MLD seasonality from float observations, (e-h) difference in the MLD between the observations and 1D GOTM) and (i-l) difference in the MLD between the observations and 3D MOM4p1 - TOPAZ. The 50m MLD contour is depicted in panels (a-d). The difference contour between observation and models of ±20 m is depicted as a solid black line in panels (e-l). The maps are 0.5°x0.5° monthly climatological medians. The difference between

observations and models is computed after the mapping. Only modeled data collocated in time and space to the floats are used to have comparable averages. The colored dots in (a) show the regions of interest characterized in Figure 8.

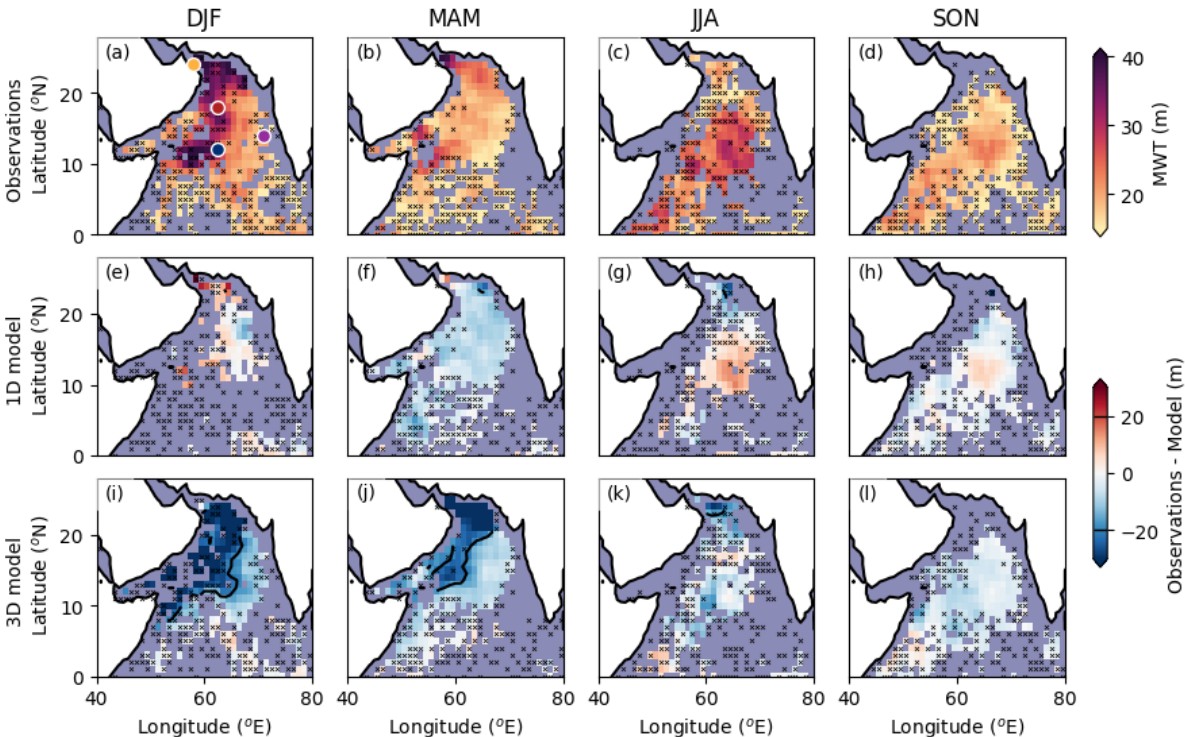

**Figure 7. MWT comparison between observations**, 1D GOTM and 3D MOM4p1 - TOPAZ. (a-d) MWT seasonality from float observations, (e-h) difference in the MWT between the observations and 1D GOTM ) and (i-l) difference in the MWT between the observations and 3D MOM4p1 - TOPAZ. The difference contour between observation and models of ±20 m is depicted as a solid black line in panels (e-l). The maps are 0.5°x0.5° monthly climatological medians. The difference between observations and models is computed after the mapping. Only modeled data collocated in time and space to the floats are used to have comparable averages. Crosses show the grid cells where less than three MWLs per season are detected in the observations. The colored dots in (a) show the regions of interest characterized in Figure 8.

### 4.4.1 Sea of Oman

The Sea of Oman (SoO) is characterized by the longest presence and thickest MWLs in the Arabian Sea (Figure 3 and 8a-c). Both 1D and 3D models reproduce the MWL's formation-erosion cycle, with some differences related to the timing of formation and boundary changes compared to the observed upper ocean stratification (Figure 8a-c).

Modeled winter mixed layers are deeper than in the observations (1D: 12±4 m and 3D 37±6m deeper; Figure 6ei and 8cd), leading to a significantly thicker modeled MWL in the 3D model (>50 m, Figures 6i and 8c). The underestimation of upper ocean stratification in this region by the 3D model is a limitation of this comparison. Despite the shift in the depth of the MLD, we believe that the pattern can show evidence of certain mechanisms driving changes in the MW cycle. After winter, both the

1D and 3D models present deeper spring-to-fall MLD (1D: 13±1 m, 3D: 16±5 m) compared to the observations (8±2 m), yet they present a more intense pycnocline, overestimating upper ocean stratification by 7% and 12% (Figure 8mn). The deepening of the upper boundary of the MWL follows this seasonal cycle: as ocean heat gain strengthens upper ocean stratification during spring and summer, the increasing density contrast inhibits vertical mixing and progressively isolates the MWL, leading to its downward migration. Despite the overestimation of the pycnocline intensity, both models reproduce this deepening trend of the MWL's upper boundary (Figure 4b, 5c, and 8bc), supporting the conclusion that upper ocean stratification buildup from heat gain is the primary driver of MWL deepening.

The observations and the modeled annual cycle differ in the timing of the formation of the MW in the region. Stratification builds up beginning of March (week 10) in both observations and the 3D model, yet there is a 1-month lag in the 1D model (3 to 5 weeks). We hypothesize that there are 3D processes that can modify MW characteristics locally and at shorter timescales by setting the formation timing. It has been shown that mesoscale eddies are likely to modulate the ASHSW altering the upper ocean stratification and the winter convective mixing (Thoppil, 2024; Trott et al., 2019). Unlike other regions of the Arabian Sea, the SoO is largely isolated from the broader monsoonal wind patterns (Chaichitehrani & Allahdadi, 2018), and its standing mesoscale eddy structures (L'Hégaret et al., 2013; Marez et al., 2019; Pous, 2004) may play a critical role in restratification timing (Font et al., 2022) or retaining the MWL, as observed in other regions (e.g., Karleskind et al., 2011; Shi et al., 2018; Xu et al., 2016).

### 4.4.2 Central Arabian Sea

In the central Arabian Sea (CAS), the upper ocean stratification and the MW formation and erosion cycle are well represented by both 1D and 3D models (Figure 8d-f). Despite the overestimation of MLD by the models in the upper ocean (observations: 39±20 m, 1D: 49±30 m, 3D: 51± 26 m; Figure 6 and 8ef), and its consequent overestimation of MWT (median MWT for observations: 16 m, 1D: 24.5 m, 3D: 36 m; Figure 7 and 8ef); the bimodal nature of MW formation in the CAS is evident. Formation timing is accurately represented, with only two weeks lag for the 1D model in winter, and on time for summer monsoon for both models (Figure 8ef). The 1D model predicts a longer MWL duration compared to observations and the 3D model, suggesting erosion is enhanced by 3-D processes. For MWs formed during winter, the observations and 3D model predict that they are eroded after May (12-15 weeks respectively, Figure 8ef), compared to the 1D model that shows MW presence until summer monsoon surface mixed layer deepening (17 weeks; Figure 8d). Similar behavior is observed and modeled for MWs formed during summer, with a duration of 5 and 3 weeks for the observations and the 3D model (Figure 8ef), and 9 weeks for the 1D model (Figure 8d). We hypothesize that in the CAS, processes captured by the 3D model—such as the southward advection of ASHSW (Han & McCreary, 2001; Prasad & Ikeda, 2002a)—might be driving the erosion of MW after both monsoons.

### 4.4.3 Eastern Arabian Sea

In the eastern Arabian Sea (EAS) advective processes play a critical role in the MW cycle. Along India's coast, a substantial
bias in MLD is observed in the 1D representation compared to the observations during winter (-38±16m, Figures 6 and 8h).
Contrary, the 3D model also presents a deep bias, but less important (-28±5 m, Figures 6 and 8i). This bias in the MLD
translates to an overestimation of the thickness of the MWL by 44% in the 1D model and 37% in the 3D model (Figure 7fj
and 8hi). During winter, freshwater from the Bay of Bengal is advected anticlockwise along India's western coast reaching up
to 20ºN (West Indian Coastal Current; Chatterjee et al., 2012) and modulating the wintertime convection there (Shankar et al.,
2016). This advection, visible in float observations and 3D model outputs, increases stratification (Figure S5gi). The haline
component of N2 changes signs in both observations and the 3D model, but it is absent in the 1D model (Figure S5h). As a
result, the 1D model underrepresents haline stratification, leading to underestimating upper ocean stratification (Figure 8m and
S5h) and overestimating convective mixing and surface mixed layers (Figure 6e-h). This bias becomes particularly noticeable
at the end of fall in the southeast Arabian Sea, when circulation shifts and overestimation of MLD is visible in the bias of the
1D model (Figure 6h).

### 4.4.3 Southern Arabian Sea

The southern Arabian Sea (SAS) exhibits a significant bias between observations and the 1D model during the summer
monsoon, indicating that a key mechanism for deep mixed layer depths and MW formation is missing in the 1D approach (red
dot in Figures 6a and 7a, Figure 8j-l). Specifically, the 1D model underestimates the MLD on average by 30% (Figure 6cg),
reducing the MWT by 31% in the SAS (Figure 7cg). We hypothesize that this bias may be due to seasonal changes in absorption
coefficients of the water (water type) that change the surface heating, which are not accounted for in the 1D model.

Despite high winds during summer, the mixed layer remains shallow in the 1D model (Figures 6a and 8k) due to enhanced
stratification caused by an overestimation of chlorophyll-induced absorption (Figure S1). Babu et al., (2004) demonstrated that
while pigment concentrations are high during July in the SAS, the MLD is deeper due to high wind speeds, leading to increased
turbulence and a deeper MLD. However, in August, as wind speeds decrease, biological heating becomes more influential
thereby overcoming the effect of wind mixing and leading to a shallower MLD. This cycle is not well-represented in our fixed
water type 1D model (Figure 8k). By reducing water turbidity in the 1D model (water type I), we better represent the observed
deep mixed layers at the start of the summer season, reducing the MLD underestimation from 32% to 14% (Figure S1).
However, we then misrepresent the shoaling and restratification that occurs when wind speeds drop, due to a lack of heat
absorption in the model (Figure S1). In contrast, the 3D model accounts for changes in water absorption properties (see Section
2.3; Morel & Antoine (1994)), providing a more accurate representation of oceanic heat absorption (Figures 6k and 8l). This
capability allows the 3D model to better capture the effects of stratification and heat distribution, highlighting the crucial role
of biological and optical processes in the SAS in regulating upper ocean stratification.

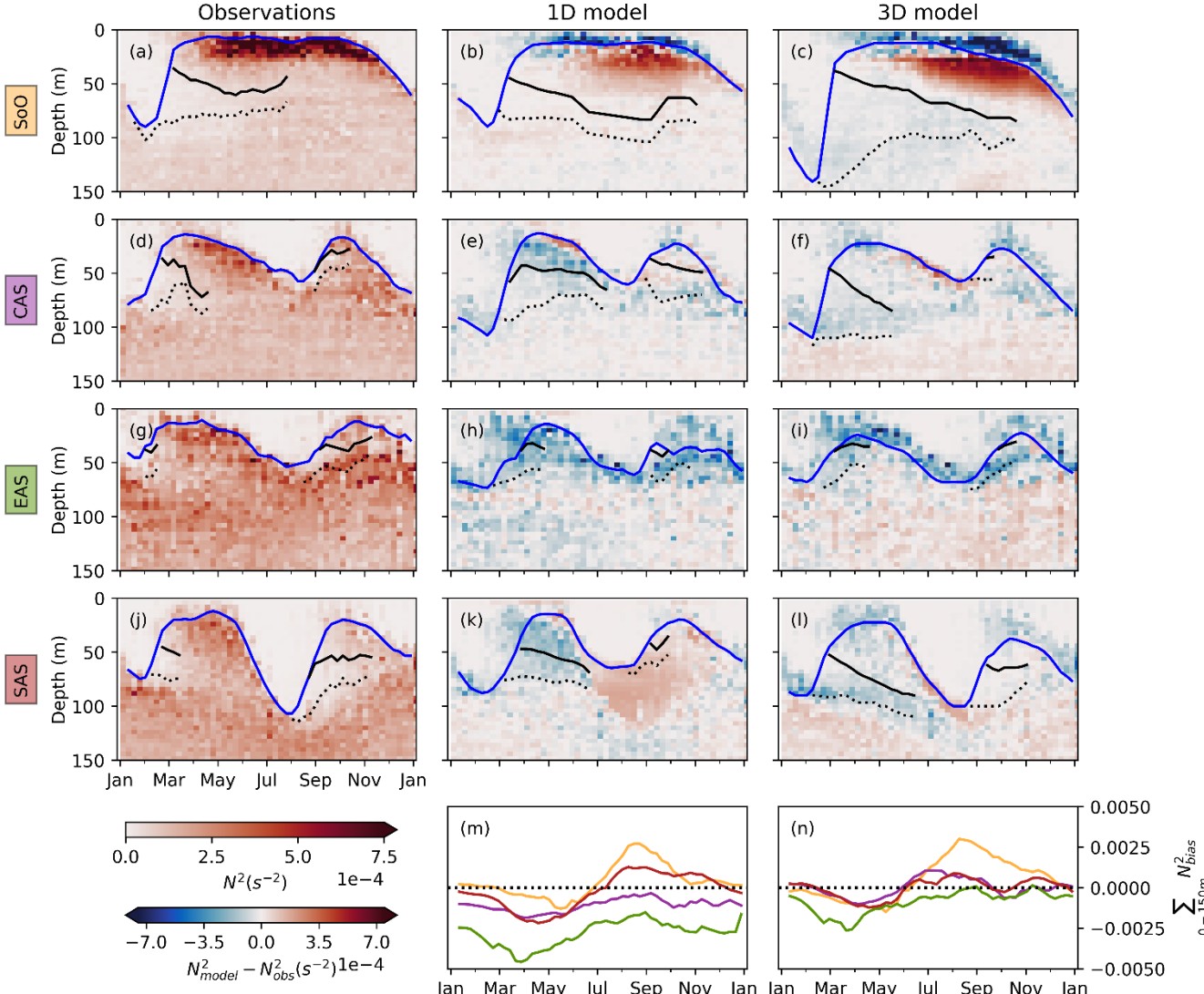

**Figure 8. Case studies - Upper Ocean Stratification from observations, 1D GOTM, and 3D MOM4p1 - TOPAZ**. (a,d,g,j) Climatological average of the annual cycle of upper ocean stratification ($N^2$) from observations and (b,e,h,k) bias between observations and 1D - GOTM and (c,f,i,l) 3D - MOM4p1 - TOPAZ model at four different case study locations: (a-c) Sea of Oman (SoO), (d-f) Eastern Arabian Sea (EAS), (g-i) Central Arabian Sea (CAS) and (j-l) Southern Arabian Sea (SAS) (See location in Figure 6a and 7a). The lines depict the average MLD (blue), the upper boundary of the MWL (solid black), and the bottom boundary of the MWL (dotted black). The upper and bottom boundaries of MWL are not shown when MWL is eroded in more than 75% of the data points used for the average. (m) 1D and (n) 3D model annual cycle of the vertical integral of the $N^2$ bias between the model and the observations for the upper 150 m of the ocean, color-coded by the different locations (SoO in yellow, CAS in purple, EAS in green and SAS in red; Figure 6a and 7a).

450

455

### 4.5 Oxygen Content of Mode Water

Mode Water's most important role is in subducting properties when formed (Lacour et al., 2023; Z. Li et al., 2023; Portela et al., 2020). Here, we quantify the upper ocean content of oxygen and MW's contribution to it. Due to a lack of observations to constrain upper ocean oxygen content and subduction and obduction rates, we use the 3D MOM4p1 - TOPAZ output to upscale the observations from floats.

MW contribution to the total oxygen content of the upper Arabian Sea (250 m) accounts annually for only 4±1% (3±1% in the southern AS, and up to 8±2% in the northern AS; not shown). We argue that the oxygen content below the mixed layer is more relevant for ventilation and biogeochemical impacts. Thus, if we only account for subsurface oxygen content (i.e. excluding surface mixed layer saturated or oversaturated waters in contact with the atmosphere), MW contribution rises to 5±1% (5±1% in the southern Arabian Sea, and up to 11±2% in the northern Arabian Sea; Figure 9b). Despite this, due to MW's intermittent presence driven by the seasonal cycle, MWs have an important contribution in certain seasons. The maximum MW oxygen content in the Arabian Sea is during spring when MWL is thickest, contributing a total of 0.030±0.007 Tmol, which is 18±6% of the total upper ocean content (Figure 9a). Splitting the contribution between the northern and the southern Arabian Sea clear differences emerge. In the northern Arabian Sea, the maximum contribution peaks to 40±17% of the oxygen content in the upper 250 m (Figure 9ad), highlighting again the importance of the presence of MW in the northern Arabian Sea in the biogeochemistry of the region. In southern Arabian Sea, during spring, MW contribution is 15±5% due to their shorter duration and thinner MWL (Figure 3, 7b, 9ad). In the northern Arabian Sea, due to their prolonged presence, MWs contribute up to 11% of the oxygen content in summer (Figure 9e), even when surface stratification is strongest, and the oxygen in the MW is completely isolated from air-sea fluxes. After the summer monsoon, MWs formed in the southern Arabian Sea are the primary contributors to oxygen content (8±3% in Arabian Sea and 9±3% in the southern Arabian Sea; Figure 9f). The remnant MWL in the northern Arabian Sea contributes to 2±1% (Figure 9f). These results showcase the importance of MW in subsurface oxygen content in a region where the anoxic waters (upper oxycline of the Arabian Sea OMZ) are close to the surface.

We acknowledge that these estimates above may overestimate the actual contribution, due to the overestimation of MWT in the model (annual average of 10% thicker than observed). Nevertheless, this provides a valuable characterization of this watermass's importance in the Arabian Sea. The MWs in the upper Arabian Sea sit just above the oxygen minimum zone (OMZ) oxycline leading to an exchange flux between the MWL and the interior, which remains unresolved by current observational or modeling approaches, to the best of our knowledge.

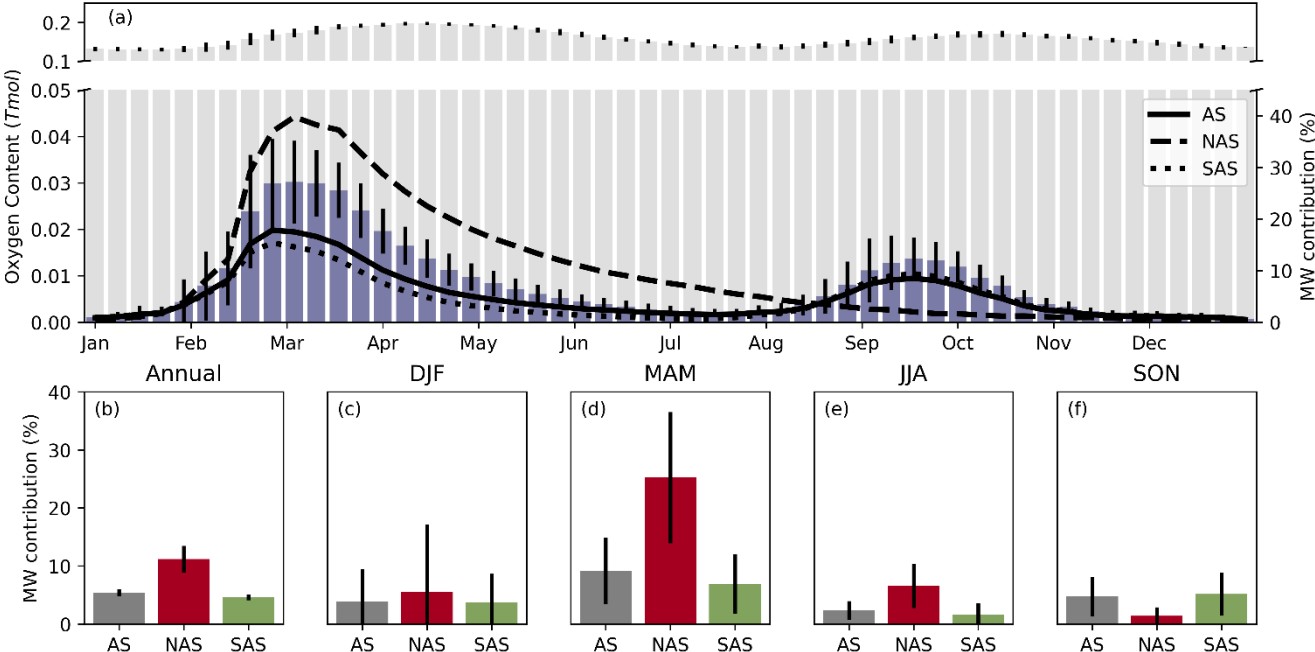

**Figure 9. Mode Water contribution to the sub-surface oxygen content of the Arabian Sea.** (a) Climatological average of the oxygen content below the surface mixed layer and up to 250 m in the Arabian Sea (gray bars) and contribution of the MWL (blue bars) from the 3D MOM4p1 - TOPAZ. Vertical bars show the standard deviation across the basin. The lines depict the percentage of contribution of MW to the total upper ocean oxygen content -excluding the surface mixed layer- for the Arabian Sea (solid line), northern Arabian Sea (dashed line), and southern Arabian Sea (dotted line). (b) Annual and (c-f) seasonal MW contribution to the oxygen content below the surface mixed layer for the Arabian Sea (gray), northern Arabian Sea (red), and southern Arabian Sea (green).

# 5 Discussion

## 5.1 Annual Cycle of Mode Water

Observational data supports the biannual subduction patterns expected in a monsoon-dominated ocean basin like the Arabian Sea (Liu et al., 2018). MW forms when surface buoyancy gain during low winds cap the deep, dense mixed layer waters beneath a surface stratified layer (Figure 4). In the northern Arabian Sea, this process occurs once a year during winter when peak annual winds occur, whereas in the southern Arabian Sea, it can occur twice a year following the monsoons (Figures 1 and 4). Liu et al. (2018) highlighted vertical pumping as the primary driver of subduction, with advection playing a secondary role. In this study, we build on this body of literature by characterizing the water mass formed through this process and determining the regional forces and mechanisms that modify MW formation and erosion. This work emphasizes that the formation of MWs is not limited to the northern Arabian Sea, nor is it restricted to the winter season.

The Arabian Sea High-Salinity Water (ASHSW) has been extensively studied as the primary MW formed in the Arabian Sea (Shenoi et al., 1993). The MW formed during the winter monsoon in the northern Arabian Sea, aligns well with the characteristics of ASHSW described in previous studies, using our climatological dataset (density range of 24-25 kg m$^{-3}$, with temperature, salinity ranges of 28–24°C, 36.7–35.3 g kg$^{-1}$, Kumar & Prasad, 1999; Figure 2). Prasad & Ikeda (2002b) describe how turbulent heat loss is responsible for the deep penetration of MLD and the convective formation of ASHSW during winter.

They argue that weaker winds, strong near-surface stratification, and weaker shear stress during the transition periods (March-May to October) account for a shallow mixed layer that caps ASHSW. We expand this body of literature by showing the formation of MWs also during summer in the southern Arabian Sea, and highlighting the importance of the interaction between buoyancy fluxes and winds in the life cycle of this water mass (Figure 4). Moreover, ASHSW is advected equatorward (Kumar & Prasad, 1999; Morrison, 1997; Prasad & Ikeda, 2002a) along the 24 kg m$^{-3}$ isopycnal surface as a subsurface salinity

maximum (Chowdary et al., 2005; Han & McCreary, 2001; Prasad & Ikeda, 2002a). While our climatological dataset does not allow us to determine the advective component towards the south, we can confirm that MW formation also occurs in the southern Arabian Sea following the winter monsoon, with MWs of a density of 24 kg m$^{-3}$ (Figure 2). Therefore, the subduction of properties occurs not only via the ASHSW in the north and its advection southward but also through direct local subduction in the southern Arabian Sea.


We have shown how MLD misrepresentation results in the misrepresentation of MW thickness and formation timing (Figures 6, 7, and 8). This emphasizes the importance of model representation of MLD. MLD is influenced by an interplay between surface buoyancy fluxes, winds and upper-ocean stratification due to circulation. Accurate representation of circulation and heat penetration is crucial for the proper simulation of both the MLD and MW life cycle in the Arabian Sea (Figures 6 and 7).

We show that monsoonal circulation is essential, as the advection of low-salinity water by the West Indian Coastal Current (October–May) acts as a stratification barrier, resulting in shallower and thinner MWLs. Without this freshwater-driven stratification, deeper mixed layers and the formation of denser waters would likely occur (Shankar et al., 2016). Moreover, freshwater stratification in the EAS has significant implications for the biogeochemistry of the region. Its presence enhances stratification and nutrient availability, yet results in thinner MWLs, reducing the buffer capacity for remineralization within

the MW. Consequently, this increases nutrient export to the OMZ and decreases physical ventilation due to stronger stratification. Moreover, we suggest a significant role of biological heat absorption in resolving upper-ocean stratification and MW formation accurately (Figure 8), as highlighted by Babu et al. (2004) for the MLD. Additionally, it has been shown that in the northern Arabian Sea, mesoscale and submesoscale eddies and frontal structures are present during the winter and can alter stratification and modulate convective mixing (Font et al., 2022; Thoppil, 2024). We hypothesize that similar to other

regions, these eddies may trap MW, preserving its properties and prolonging the presence of this buffer layer (e.g., Karleskind et al., 2011; Shi et al., 2018; Xu et al., 2016). Therefore, further investigation into how eddies and small-scale frontal structures in the Arabian Sea influence MW properties, as well as their impact on the ecosystem, is essential. High-resolution, eddy-

resolving models and targeted high-resolution observations are needed to assess their effects on stratification, MW formation, and length of MW presence until full erosion.


## 5.2 Implications of Mode Water for Heat and Oxygen Fluxes in the Arabian Sea

The MWL acts as a barrier between the atmosphere and the interior, impacting the Arabian Sea stratification and mixing processes and thus modulating heat and oxygen exchange. We argue that MW's presence can store heat below warm surface mixed layers. MWL does provide a buffer of warmer waters compared to the interior, therefore the cooling potential of entrainment is reduced with this barrier layer. MW presence could maintain the Arabian Sea Warm Pool by effectively inhibiting the vertical mixing of the upper ocean with deeper cooler waters. This process has been previously described regarding barriers layers formed by freshwater advection from the Bay of Bengal in the southeast Arabian Sea (Echols et al., 2020; Li et al., 2023), contributing to the genesis of surface intensified marine heat waves (Saranya et al., 2022). Moreover, the interaction between freshwater Bay of Bengal barrier layers and MW is crucial and warrants further investigation. The extent to which barrier layers influence MW away from the coast remains uncertain, highlighting the need for targeted studies to better understand its geographical constraints and impacts.

Mode waters, both locally and remotely formed (e.g., Subtropical Underwater, Indian Central Water, and Subantarctic Mode Water) influence oxygen distribution in the Indian Ocean (Ditkovsky et al., 2023). Here, we focus specifically on the locally formed Arabian Sea Mode Water and its potential impacts on regional oxygen dynamics. MW sits on top of the upper oxycline of the Arabian Sea OMZ. The presence of the MWL acts to limit the transfer of kinetic energy from the atmosphere to the oxycline, reducing ventilation; at the same time, it provides an oxic layer for remineralization of organic matter thereby diminishing biological oxygen demand deeper in the OMZ (Weber & Bianchi, 2020). As such, MW reduces the coupling of the OMZ to surface processes driving oxygen supply to or demand in the OMZ. We show that MW, whether its subduction is seasonal or long-term (Liu et al., 2018; Zhou et al., 2023), has large importance and seasonality on upper ocean oxygen content in the region, and thus its presence is unarguably crucial regarding changes the boundaries of the OMZ (Figure 9). We hypothesize that MW acts as a more important oxygen reservoir in the northern Arabian Sea than in the south, and specifically in the Sea of Oman (Figures 3p and 9), where its volume and residence time are largest and thus provide more time and volume for remineralization of sinking matter to occur. The extended presence of MW in the SoO suggests that it can be an especially important layer to sustain remineralization, thereby reducing biological oxygen demand in the core of the OMZ, and providing oxygen locally to the upper OMZ via diapycnal processes across the oxycline. Various processes that mix and stratify capped surface waters need to be explored further, such as sub-mesoscale fronts within the MWL, eddies, topographical interactions, internal waves, and biophysical interactions. The timescales over which this oxygen dissipates into the ocean's interior or is remineralized remain unexplored. Further work is needed to study the fate of oxygen in the MWL at high-resolution and ecosystem impacts.

Ditkovsky et al. (2023) showed strong trends of deoxygenation for the boundaries of OMZ. Increases in stratification inhibit local ventilation and promote deoxygenation (Schmidtko et al., 2017), yet other studies have suggested that increases in stratification will cause a reduction in productivity, and ultimately lead to a decrease in DO consumption by respiration (Bopp et al., 2013; Kwiatkowski et al., 2020). MW can be relatively important in driving trends of the boundary of OMZ which presents strong deoxygenation both from biogeochemical buffering on respiration and direct physical oxygen supply. Thus, we believe that accurate representation of MWs in climate models is crucial due to their importance in both physics and biology and their fine interplay. These require further modeling and observation, with targeted studies quantifying vertical fluxes of oxygen, carbon, and nutrients in OMZ and through MWL.

### 5.3 The Future of Mode Water in the Arabian Sea

Climate change is expected to modify upper ocean stratification and thus MW characteristics and its impact on the broader ecosystem. In recent decades, SST, stratification, and upper ocean heat content show increasing trends, indicating that the Arabian Sea has been warming and that the stratification has strengthened (Albert et al., 2023; Mohan et al., 2021; Nisha et al., 2024), also showing clear warming trends in all water masses in the Arabian Sea (Shee et al., 2023). These changes have a direct impact on the broader ecosystem, with for instance net primary productivity decrease in Arabian Sea in the recent decade (Maishal, 2024). Interannual and climate variability (e.g. Indian Ocean Dipole) in atmospheric forcing and ocean circulation are expected to affect the formation, erosion, and characteristics of MW. The changes in the MW life cycle at these timescales warrant further investigation. Moreover, over multidecadal timescales, the weakening of the monsoonal winds over the Arabian Sea as well as changes in precipitation rates are predicted to continue (Cheng et al., 2025; Luo et al., 2024; Parvathi et al., 2017). We have demonstrated that the MW life cycle and thickness are very sensitive to the previously mentioned atmospheric forcing, namely heat and freshwater fluxes and wind intensity (Figure 4). Weakening of winds has been shown to reduce winter mixed layer deepening in the northern Arabian Sea and hence less oceanic productivity (Lachkar et al., 2018; Parvathi et al., 2017). As per the tight relationship between MLD and MWT and presence, we hypothesize that the weakening of winter convective mixing will result in thinner MWL and less buffering capability for remineralization. Moreover, freshening of surface waters due to increased precipitation is anticipated (Sharma et al., 2023), which will likely lead to changes in upper ocean stratification and subsequently affect MW formation - we have shown how salinity can change MW volume. There is a wind vs buoyancy flux balance that defines the latitudinal bimodality on the monsoonal formation of MW. With climate change and associated changes in buoyancy fluxes and wind mixing, the annual to biannual latitudinal boundary is going to shift. Moreover, we showed the tight interaction between buoyancy and winds in setting the formation of MW during the summer monsoon. If the ratio changes between these forcings, it could result in a fast blocking of subduction in the south. These changes have undeniable importance in Arabian Sea's physical and biogeochemical functioning. For instance, changes in the subduction capacity of MW could influence directly ventilation from 1 to 2 MWLs annually and could have big

implications for instance for the depth of the upper oxycline of the OMZ or the southern extent of the OMZ. Thus, we must investigate the response of Arabian Sea MW and its impacts on the region's physics and biogeochemistry in future climate scenarios.

## 6 Conclusions

In the Arabian Sea, MW formation is first driven by atmospheric forcing and monsoons, then regional variability influenced by circulation patterns and spatial monsoon variability, which exhibit significant seasonal variations. MW formation occurs in two distinct temporal modes with yearly production in the northern Arabian Sea and biannually in the southern Arabian Sea, following the summer and winter monsoons. MWL characteristics are first order determined by buoyancy and wind mixing processes. Heat loss during winter monsoon forms deep mixed layers in the Arabian Sea that are capped by the formation of a shallow stratification when the ocean gains buoyancy after the winter monsoons. This water layer erodes over time and has the potential to mix its properties with the ocean interior. During summer monsoon, buoyancy loss and strong winds from deep mixed layers in the southern Arabian Sea are also capped when winds reduce and surface heating stratifies the upper ocean.

Regionally, the MW life cycle is influenced by the advection of water masses that change the stratification of the upper ocean. For instance, in the eastern Arabian Sea, the advection of freshwater-driven stratification from the Bay of Bengal by the West Indian Coastal Current can limit the depth of surface mixed layers and thus precondition the MW formation and characteristics. Another factor that determines the MWL is the seasonal cycle of optical properties of the water, with the most important in the southern Arabian Sea, where in summer biological heating determines the depth of the summer surface mixed layers preceding the formation of MW. We highlight the importance of accurately resolving the surface mixed layer (e.g. biological heating, advection) for a good representation of the MWL. The observed annual maximum volume of MW is $6.3 \times 10^{13}$ m$^3$, with a larger importance for upper ocean volume and oxygen content in the northern Arabian Sea due to thicker MWL and longer duration of its presence. We estimate the contribution of MW to the oxygen content of the upper ocean to be $4 \pm 1\%$ in the Arabian Sea, but due to its intermittent presence driven by the seasonal cycle, MW contribution in certain periods peaks to $30 \pm 6\%$ of the oxygen content of the upper 250m in the northern Arabian Sea during spring. Our results highlight the importance of MW as a barrier layer for the physics and biogeochemistry of the Arabian Sea.

**Code availability** The software associated with this manuscript is published on GitHub https://github.com/EstelFont/Mode_Water_Arabian_Sea.

**Data availability.** The Argo data used in this study is available by the International Argo Program and the national programs contributing to it for the domain 40–80 °E, -5–30 °N and between 2000-2023 (Argo, 2023). The MOM4p1-TOPAZ potential

temperature, salinity, and oxygen data used in this study for the upper 250m and domain 35-80 °E, 0-30 °N, are available from Zenodo (Font & Vinaychandran, 2025). Bathymetry data is available by GEBCO Compilation Group (GEBCO, 2023).

**Author contribution:** EF, BYQ, and SS conceptualized the study. EF performed the data curation and formal analysis and wrote the manuscript draft. BYQ, SS, and PNV reviewed and edited the manuscript.

**Competing interests:** The authors declare that they have no conflict of interest.

**Acknowledgements:** The MOM4p1-TOPAZ runs were carried out on the Param Pravega supercomputer at SERC, Indian Institute of Science, Bengaluru. Thanks to GFDL for providing the source code of the coupled physical–ecosystem model (MOM4P1-TOPAZ). The authors thank P. Neema for subsetting the MOM4p1-TOPAZ dataset.

**Financial Support:** EF and BYQ are supported by ONR GLOBAL Grant N62909-21-1-2008 and by Formas Grant 2022-01536. EF and SS are supported by a Wallenberg Academy Fellowship (WAF, 2015.0186) and by the Swedish Research Council (VR, 2019-04400). S.S. has received funding from the European Union's Horizon Europe ERC Synergy Grant program under grant agreement No. 101118693 (WHIRLS). PNV acknowledges support from the BRICS-PARADIGM project, Department of Science and Technology, Govt. of India.

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
