# Peer review of "On Mode Water formation and erosion in the Arabian Sea: Forcing mechanisms, regionality, and seasonality"

_EGUsphere, 2025_

## Author Comment (AC1)

Author response to comments on "On Mode Water formation and erosion in the Arabian Sea: Forcing mechanisms, regionality, and seasonality" by Font et al., (Preprint egusphere-2025-468)

We thank the referee for critically reading this manuscript and providing helpful feedback, which has added a great deal to improve the manuscript and clarify certain sections.

We respond to all issues addressed in their comments below, as well as we add the revised changes in the manuscript. The Reviewer comments are included here in black, and the answers below the respective comments in blue. The text that has been modified in the manuscript according to the reviews is presented in *italic*. The line numbers in the answers refer to the new manuscript version after the suggested changes.

**Referee #1**

this paper is clear and well written; it is novel and useful to the community; the first figure is pedagogical

I recommend publication after addressing a few minor comments

1) what is the influence of the vertical mixing coefficient in the models, on the results?

The vertical mixing coefficient is crucial for stratification and significantly influences the model results. The selected mixing scheme for the 1D model—KPP (K-profile parameterization)—was chosen because it covers a wide range of open ocean conditions and allows for direct comparison and compatibility with the mixing scheme used in the 3D model (see line 126 in Section 2.3). Previous studies have evaluated the most suitable mixing schemes for Indian Ocean dynamics (Tirodkar et al., 2022), with a particular focus on KPP and K-epsilon mixing schemes. While K-epsilon performed slightly better and has been successfully applied in the Sea of Oman (Khalilabadi, 2024), we use KPP in our model to ensure consistency between the 1D and 3D configurations, and because it provides a robust representation of observations.

A detailed assessment of individual vertical mixing processes would require a dedicated study with a more refined approach. However, for our objective—characterizing seasonality and regional variability—the model successfully captures the observed cycle, giving us confidence that the chosen coefficients provide a realistic and robust representation within the model framework.

We have specified in the text that GOTM uses KPP mixing scheme *"... using the KPP mixing scheme (K-profile parameterization; Large et al., 1994)"*

*Tirodkar, S., Murtugudde, R., Behera, M. R., & Balasubramanian, S. (2022). A comparative study of vertical mixing schemes in modeling the Bay of Bengal dynamics. Earth and Space Science, 9, e2022EA002327. https://doi.org/10.1029/2022EA002327*

*Khalilabadi, M. Turbulent Processes in the Oman Sea: A Numerical Study. Water Resour 51, 98–109 (2024). https://doi.org/10.1134/S0097807823600717*

2) in a 1D hydrological (large-scale) model, potential vorticity PV will be directly related to $N^2$ but in a 3D model it has dynamical components; they may be of use in characterizing the MW in the 3D model simulations ; have you evaluated PV in the 3D model in the MWL - and its gradients above and below ?

We appreciate the reviewer's suggestion regarding potential vorticity (PV) as a diagnostic for the MWL in the 3D model. We initially explored this approach but found that differences in data sources, mainly due to variations in vertical and temporal resolution, resulted in noisy PV fields, making it challenging to use as a consistent diagnostic. Given these limitations, we opted for a density threshold approach that ensures consistency across datasets. Careful assessment of the density thresholds over the various datasets has given us confidence that this metric is well suited to define MW boundaries (Section 3.1).

That said, we agree that analyzing PV gradients in the 3D model could provide valuable insights into MWL structure and evolution. A future study focused solely on 3D model output could refine this approach, minimizing inconsistencies introduced by mixed datasets and improving MW detection. We appreciate this insightful suggestion and will consider it in future work.

3) concerning the difference between 1D and 3D results (fig.5-6) it may be of interest to plot MKE and EKE next to these figures to assess the local importance of horizontal advection

We agree that considering mean kinetic energy (MKE) and eddy kinetic energy (EKE) can help assess the role of horizontal advection in explaining the differences between the 1D and 3D results. Rather than adding these fields to the figures, MKE and EKE have been extensively studied in the region, partly because fieldwork difficulties have emphasised the importance of remote sensing studies in the region. We have included two citations (one specifying the figure we refer to, *Figure 1 in Zhan et al., 2020*), highlighting regions where MKE and EKE are particularly relevant. Additionally, we have expanded the discussion to clarify how horizontal advection plays a locally significant role and contributes to the observed biases:

Line 332-335: *"horizontal advection, particularly in high eddy kinetic energy regions, redistributes heat and momentum laterally, a process absent in the 1D model. In the western Arabian Sea, where eddy kinetic energy intensifies in summer near the Somalian coast (Figure 1 in Zhan et al., 2020; Sun et al., 2022), this likely contributes to the MLD differences between the 1D and 3D simulations (Figure 6c,g,k)."*

We believe this addition sufficiently addresses the reviewer's concern while maintaining clarity in the figures.

*Zhan, P., Guo, D., & Hoteit, I. (2020). Eddy-Induced Transport and Kinetic Energy Budget in the Arabian Sea. Geophysical Research Letters, 47(23), e2020GL090490. https://doi.org/10.1029/2020GL090490*

*Sun, C., Zhang, A., Jin, B., Wang, X., Zhang, X., & Zhang, L. (2022). Seasonal variability of eddy kinetic energy in the north Indian Ocean. Frontiers in Marine Science, 9, 1032699. https://doi.org/10.3389/FMARS.2022.1032699/BIBTEX*

4) in your equation (1) you merge the thermal and haline contributions to buoyancy ; what are their respective roles in MW formation and disappearance ; how uniform are T and S when rho is constantvertically ?

Indeed, this is a good suggestion. We have assessed the relative contributions of the thermal and haline components to the total buoyancy flux. The thermal component is two orders of magnitude larger, accounting for 99.5% of the buoyancy flux on average. We have added the following statement:

Line 248:"*The buoyancy flux is driven by its thermal component ( (α·Q_NET)/(ρ_0·c_p )) ≫ β·S_A·(E-P), Figure S2)*".

We have also updated Figure S2 in the supplementary material to include the contributions of each component.

[Figure]

To address the reviewer's point about T-S variability within the MW layer, we computed the standard deviation of temperature and salinity across the MWL. Both axes of the standard deviation plot are scaled by α/β=0.34 to reflect equal contributions to density changes (Δρ=10−4 kg m−3). Our results indicate that temperature variability within the MWL is larger

than salinity variability in terms of their respective contributions to density, but overall, variations in T-S within the constant-density layer remain minor.

[Figure]

5) you mention barrier layers in relation with BBW; this latter flows along the eastern boundary of the AS ; how extended geographically (away from the coast) is the influence of these barrier layers on MW ?

The role of barrier layers formed by freshwater advection from the Bay of Bengal in the southeast Arabian Sea has been documented (Echols et al., 2020; Li et al., 2023) in recent studies, although questions remain about how far offshore barrier layer influence on MW remains extends. We feel that properly addressing this question, including understanding the seasonal variations and the drivers of variability of its offshore influence, would constitute an entirely separate study. To acknowledge this open question, we have added the following statement:

Line 546-551: *"This process has been previously described regarding barrier layers formed by freshwater advection from the Bay of Bengal in the southeast Arabian Sea (Echols et al., 2020; Li et al., 2023), contributing to the genesis of surface intensified marine heat waves (Saranya et al., 2022). Moreover, the interaction between freshwater Bay of Bengal barrier layers and MW is crucial and warrants further investigation. The extent to which barrier layers influence MW away from the coast remains uncertain, highlighting the need for targeted studies to better understand its geographical constraints and impacts."*

*Echols, R., & Riser, S. C. (2020). The Impact of Barrier Layers on Arabian Sea Surface Temperature Variability. Geophysical Research Letters, 47(3), e2019GL085290. https://doi.org/10.1029/2019GL085290*

---

## Author Comment (AC2)

Author response to comments on "On Mode Water formation and erosion in the Arabian Sea: Forcing mechanisms, regionality, and seasonality" by Font et al., (Preprint egusphere-2025-468)

We thank the referee for critically reading this manuscript and providing helpful feedback, which has added a great deal to improve the manuscript and clarify certain sections.

We respond to all issues addressed in their comments below, as well as we add the revised changes in the manuscript. The Reviewer comments are included here in black, and the answers below the respective comments in blue. The text that has been modified in the manuscript according to the reviews is presented in *italic*. The line numbers in the answers refer to the new manuscript version after the suggested changes.

**Referee #2**

This paper uses Argo observations alongside one- and three-dimensional models to examine the formation mechanisms of Arabian Sea High Salinity Water, one of the key water masses in the upper Indian Ocean. ASHSW formation is found to be atmospherically driven in the main, although there are regions where ocean processes exert an appreciable influence.

The formation of ASHSW received a fair bit of attention a few decades ago: it is nice to see the topic revisited with new observations and tools, and a fresh approach. I have a few comments and questions below, but this is a well-written paper with nice figures, and I suspect that the revisions I've suggested are fairly minor.

Line 27. "… Over the Argo era…" – give the year, just for clarity?

Thank you. We added to the period. Text changed to: Line 27 "*... over the Argo era (2005–2020, Z. Li et al., 2023)."*

Line 39. The paper has a lot of acronyms in it: I might not use one for Arabian Sea.

We agree with the referee on reducing the number of acronyms. The Arabian Sea is not an acronym anymore.

Line 42. Comma after mass?

Thank you, added.

Line 62. Acronym already defined, so should be OMZ.

Thank you, changed.

Line 68. My feeling is that this sentence should be the start of a new paragraph – perhaps joined with the one after?

Thank you for the suggestion. We moved it to the start of the following paragraph to improve the flow of the text (see line 72).

Line 85. The number of profiles for half-degree bin: is this over the whole time period?

Yes, it is over the whole period. We have clarified it in the text (line 85: "*over the studied period*" and in the caption of Figure 1: "*The total*")

Line 145. Is mode water thickness defined on depth or density? The upper and lower boundaries are defined in density – but is this then converted into a depth? Either is fine, but would be good to state clearly.

The MWT is defined in depth space. We have clarified (Line 149. "*on either side of the core of the MWL in depth space*"

Line 147. Have the authors done any sensitivity analysis on these definitions? For instance, does choosing 30 days give very different results to choosing 20? Or 40?

Thank you for your comment regarding the sensitivity of our definitions. The choice of 10 m for MWT is constrained by the vertical resolution of our data gridding, which is 4 m. To ensure robustness in our analysis, we require at least three data points within the MWL. Defining MWT with a thinner threshold (e.g., 8 m) could lead to increased uncertainty, while a thicker threshold (e.g., 12 m) would not substantially alter our conclusions. The 30-day duration was selected as an optimal balance to minimize the influence of data gaps, given that our dataset has a 10-day temporal resolution. A shorter threshold (e.g., 20 days) might be more sensitive to transient variations, while a longer threshold (e.g., 40 days) could smooth out meaningful variability. The stratification threshold of $5 \times 10^{-5} s^{-2}$ was chosen to represent a physically meaningful transition, as it approximates the background stratification typically observed in the region outside of the MWL. This ensures that the erosion definition captures the point at which the MWL is no longer distinguishable from the surrounding water column. To clarify these choices, we have now explicitly stated this rationale in the revised manuscript as follows:

Line 158-162: "*The erosion time of the MWL is defined as when the stratification at the core of the MWL exceeds $5 \times 10^{-5} s^{-2}$ ($N^2_{MWL} > 5 \times 10^{-5} s^{-2}$) or when its thickness falls below 10 m (MWT < 10 m), sustained for at least 30 days. The $N^2_{MWL}$ threshold represents the approximate background stratification in the region, ensuring that MWL erosion is defined relative to physically meaningful conditions. The 10 m threshold ensures sufficient vertical data coverage given the 4 m depth resolution, while the 30-day duration is chosen to account for the 10-day temporal resolution and minimize the impact of data gaps*."

We hope this addresses your concern.

Line 172. I think the paper would be clearer if the authors referred to the monsoons either as summer and winter, or north-east and south-west.

We now refer solely to summer and winter monsoons and have removed all the NE-SW monsoon references.

Line 182. "Equatorial"?

Thank you, changed.

Line 185. "It's"?

Thank you, changed.

Line 191. "Monsoons"?

Thank you, changed.

Line 192. I'm not quite sure what this sentence is trying to say – and especially because "the importance of the North" has not yet been demonstrated.

Apologies for the confusion. We have removed the sentence as it was unclear, and we believe did not bring any information that is included later in the paragraph.

Line 195. Is there a typo in this equation? ImpactFactorregion appears on both sides and so, as written, should cancel… Also, what does region mean? I assume northern and southern Arabian Sea but, as written, it's not crystal clear. Also, if ImpactFactor is volume over area, could it be described as a scale depth of the mode water later, perhaps?

Thank you. There was a typo in the equation, and the right-hand side Impact Factor had to be MW volume. We have fixed the typo and restructured the paragraph and definition of this metric to be clearer and precise.

Line 210-215: "*Thus, we build an impact factor scaling the volume of MW by the contribution of the northern and southern Arabian Sea to the total Arabian Sea area (Figure 3p). The impact factor of MW is computed as Impact Factor_region= MW volume_region·(Area_Arabian Sea∕ Area_region); where the MW volume for each grid cell is computed as the MWT multiplied by the cell area; the Arabian Sea area is defined by the ocean area shown in Figure 3a and the region areas are the northern or the southern Arabian Sea delimited by the 20ºN parallel. The northern Arabian Sea is 10% and the southern Arabian Sea the 90% of the total area.*" and Caption in Figure 2 "*...as MW volume scaled by the region's area contribution to the total Arabian Sea area*".

Moreover, there was also a typo in the units as the Impact Factor is a volume. Thus, we updated the units in Figure 3p.

Line 199. This sentence makes no sense.

Apologies for lack of clarity. We have paraphrased it to Line 215-216: "*The impact of the northern Arabian Sea on the total volume of MW is larger than the south in the first half of the year, despite its area being 10% of the total Arabian Sea (Figure 3o).* "

Figure 3. Panels (o) and (p) – say north and south, to match the names for the regions used in the text (if I've understood the text correctly), rather than re-stating the definition. And plotting things against day/week or year is one of my personal bug-bears – plotting against month is much more straightforward. (See also Figures 4, 8 and 9.)

We changed the labels in panels o and p to north and south to match the text and modified the Figure caption accordingly. Also, we agree that changing the x-axis to month instead of

day/week improves readability of the figures. Figures 3,4,8,9 have now per month on the x axis.

Line 226. Again, this sentence makes no sense. I have turned the phrase "latitudinal regionality" over and over, but I cannot understand it.

Apologies, we agree it lacks clarity. We have paraphrased to Lines 244-246: "*The annual cycle of MW presence, represented as the percentage of float profiles containing MW per latitudinal bands, shows the formation once a year in the northern Arabian Sea and the biannual presence in the southern Arabian Sea (Figure 4a).*"

Line 228. "collocated": co-located? Does this involve interpolation, or do the authors just pick the closest ERA5 grid point?

Thank you for this comment. We agree that it is important to clarify how the ERA5 is collocated. We have chosen the closest ERA5 grid point, and now it is specified changing collocated to Line 247: "*...from ERA5 closest to the float profiles..* ".

Line 270. Seasons haven't been defined? Does this mean "per three month period"?

Thank you, we acknowledge the importance of defining the seasons, We have specified in the methods section Line 162-163: "*Throughout this study, we use the definition of seasons based on three-month periods: winter monsoon (December–February, DJF), spring (March–May, MAM), summer monsoon (June–August, JJA), and fall (September–November, SON).*".

Line 298. "… simplifying dynamics to their maximum". This is a pretty vague phrase – might be good to be a little more specific?

Thank you, we have changed this sentence to be clearer and specific to Lines 317-318: "*we can represent this layer by isolating vertical processes (1-D) from the three-dimensional (3-D) dynamics.*"

Line 306. I'm not sure that I understand the logic here: surely the existence of a regionally invariant bias would suggest limitations in the ability of a one-dimensional model to replicate mixed-layer processes?

We appreciate the reviewer's comment and the opportunity to clarify our statement. Our original intent was to convey that the regional variability in the 1D model MLD bias suggests that the model's ability to represent mixed-layer processes is influenced by local dynamics, which are not uniformly captured by a purely 1D approach. To improve clarity, we have reworded the sentence as follows: Lines 327-328 "*The 1D model MLD bias displays regional variability, which could indicate that the 1D approach cannot fully represent surface mixed layer processes.*" This revision aims to better reflect our reasoning while addressing the reviewer's concern. We hope this modification resolves the ambiguity.

Line 362. Modulate ASHSW how?

Thank you. We have clarified and specified how it could change ASHSW by altering the upper ocean stratification and the winter convective mixing.

Line 452. I'm not sure reminiscence is what you mean here?

Thank you. We changed it to "*The remnant MWL*"

Line 529. More important than what?

Thank you. We have clarified the comparison. Line 562 "*...as a more important oxygen reservoir in the northern Arabian Sea than in the south,*"

Line 544. ESM – acronym not defined? Also I'm pretty sure "impotence" is a typo…?!

Thank you. Removed the acronym to the earth system model and substituted to climate models. We appreciate the referee's thoroughness and apologize for the typo. We have changed to the correct word: importance. See line 577: "*...MWs in climate models is crucial due to their importance….*"

---

## Author Comment (AC3)

Author response to comments on "On Mode Water formation and erosion in the Arabian Sea: Forcing mechanisms, regionality, and seasonality" by Font et al., (Preprint egusphere-2025-468)

We thank the referee for critically reading this manuscript and providing helpful feedback, which has added a great deal to improve the manuscript and clarify certain sections.

We respond to all issues addressed in their comments below, as well as we add the revised changes in the manuscript. The Reviewer comments are included here in black, and the answers below the respective comments in blue. The text that has been modified in the manuscript according to the reviews is presented in *italic*. The line numbers in the answers refer to the new manuscript version after the suggested changes.

**Referee #3**

Local Mode Water dynamics are critical for understanding the physical oceanography (and thus the biogeochemical oceanography) of the Arabian Sea. The understanding of these dynamics contributed by the manuscript is valuable for understanding ventilation of the Arabian Sea OMZ, and will help in evaluating future ocean and climate model representations of Arabian Sea ventilation. The manuscript highlights significant and bimodal contributions of the southern Arabian Sea to MW formation, when previous literature has focused on northern winter MW formation.

The analysis performed in this study is thorough, and draws upon the relevant physical frameworks for analyzing changes in stratification related to Mode Water Formation. The use of both observations and models is a strength of the manuscript. I believe that the comparison of the 1D and 3D model to infer the importance of horizontal dynamics is sound, though I have some questions about the role of turbidity in the comparison.

The takeaway messages are clear, but some sections could be more concise or clearly structured, and some figures more clearly introduced.

I recommend minor revisions, with specific points listed below. In the points below, I also include a one or two new pieces of analysis that I would be interested in seeing. However, the manuscript already contains sufficient results, and no new results are necessary for the manuscript to be fit for publication. Overall, I think that the manuscript makes valuable contributions to the topic.

Specific points:

Lines 135-145: To clarify, this algorithm is strictly one-dimensional, correct? So if a MWL exists at a given point, but was formed elsewhere, it would not appear in e.g. Figure 3c-n? Do you think that this may lead to cases where a MWL seems erroneously eroded, if it remains intact but advected to a location that did not have a MW formation event?

The MW detection algorithm is indeed strictly one-dimensional. However, since it is based on a density threshold, it allows for the identification of low-stratified water that has been advected from other regions. This means that while the algorithm does not explicitly track

advection, MWL that has formed elsewhere and subsequently moved into a given location can still be detected, provided it meets the density criteria.

That said, we acknowledge that in cases where advection leads to MWL modification or mixing, there could be instances where the detected MWL appears thinner or more eroded than it actually is. We will clarify this point in the text to address potential misinterpretations. Lines 151-153: "*While the algorithm does not explicitly track advection, the density threshold allows for the detection of low-stratified MWL that has formed elsewhere and been advected into the region.*"

Lines 144-145: In Figure 1, sometimes the MWD is above the bottom of the MWL, including in the schematic in Figure 1b. So, the physical intuition for these metrics is a bit fuzzy. Can you clarify this? How is it that the Mode Water layer can extend deeper than the deepest surface mixed layer? Is the 0.05 kg/m3 threshold universally justified, or does it break down in some regions?

We thank the reviewer for this comment and agree that further clarification is needed. The MWL can extend deeper than the deepest surface mixed layer due to the advection of denser MW formed in other regions, which can result in a thicker low-stratified layer at a given location. This is one of the reasons why we use a density threshold criterion. We clarified this point (see response to previous comment)

Regarding the choice of the 0.05 kg/m³ density threshold, we tested multiple criteria and found that this value provided the best balance between sensitivity to both Argo observations and model outputs, making it the most robust and universal choice across the basin. However, we acknowledge that regional variations may exist, but this threshold represents the best compromise between sensitivity and broad applicability across the basin.

Results 4.1: this section is quite hard to follow because the text jumps between Figures 2,3 and 4 without actually introducing any of them. I think that Figure 2 is a clear and simple starting point for contrasting the North, Central and South Arabian Sea, but it doesn't really get any attention or explanation. Also perhaps a separate color scheme for the samples in each section would highlight the regional contrast.

We acknowledge that the flow of this section could be improved by better introducing the figures and providing a clearer structure. To address this, we have made the following revisions:

- Revised the introduction of Figures 2, 3, and 4 – We now introduce Figure 2 at the beginning of Section 4.1 as the primary reference for contrasting the North, Central, and South Arabian Sea before discussing Figures 3 and 4. We have revised the text as follows: Lines 175-179: "*In this section, we characterize the timing of formation, erosion, duration, and thickness for the contrasting monsoons and regions. Figure 2 provides an overview of the MW T-S properties colored by latitude, serving as the basis for contrasting the North, Central, and South Arabian Sea. Building on this, Figure 3 further illustrates the duration of the MWL, MWT, and the seasonality of the MW volume, while Figure 4 highlights MW formation relation to atmospheric forcing.*"

- Updated the color scheme – We have implemented a segmented colorbar every 5 degrees instead of a continuous colorbar to highlight latitudinal contrast in Mode Water T-S properties.

[Figure]

These adjustments should improve readability and ensure a clearer narrative for the reader. We appreciate your suggestion and believe these changes enhance the clarity of our results.

Lines 303-310: May strengthen your argument to touch upon some things that the models represent well, in addition to biases. For example, the 3D model capturing the summer mixed layer depth

We thank the referee and have now modified the text to add this in Lines 329-332 "*The MLD bias is consistent across the basin during winter, posing a constraint to our analysis. However, as the seasonal cycle is well reproduced, and the 3D model accurately captures the deep summer mixed layer in the center of the basin, we consider the model suitable for assessing MW life cycle mechanisms*."

Lines 318-319: This is a really nice point. MLD is much easier to diagnose in models than the Mode Waters themselves, and I would be interested in seeing if any more quantitative connections can be made. For example, you could look at a few timeseries for different regions to show how the MLD biases lag the MWT biases and how the magnitudes compare. Is it pretty consistent? Does it change by region?

We thank the reviewer for raising this important point. While a detailed investigation is beyond the scope of this study, we acknowledge that it warrants further attention, particularly from a modeling perspective. We have added a sentence emphasizing the significance of this metric and the open questions that remain following our analysis in Lines 344-346 "*The relationship between MLD and MW biases warrants further investigation. As MLD is easier to diagnose in models, analyzing its regional biases and lagged relation to MWT biases could help refine model representation of this layer and better constrain MW ecological implications*."

Line 322: "major differences and various dynamics" reads a bit clunkily here

We have removed it to make it more straightforward, as it wasn't bringing much content.

Lines 356-357: I'm missing the connections here that lead to the takeaway that the MWs are "driven by the build-up of upper ocean stratification during spring-summer from ocean heat gain". The reasoning in this paragraph is a bit unclear

Thank you for your comment. We recognize that the reasoning connecting upper ocean stratification during spring-summer to MWL deepening was not explicitly stated in the original text. To clarify this, we have revised the paragraph to make the logical connections more explicit. Specifically, we now explain how ocean heat gain strengthens stratification, which increases the density contrast, inhibits vertical mixing, and progressively isolates the MWL, leading to its downward migration. The revised text at the end of the paragraph now reads: Lines 381-386 *"...The deepening of the upper boundary of the MWL follows this seasonal cycle: as ocean heat gain strengthens upper ocean stratification during spring and summer, the increasing density contrast inhibits vertical mixing and progressively isolates the MWL, leading to its downward migration. Despite the overestimation of the pycnocline intensity, both models reproduce this deepening trend of the MWL's upper boundary (Figure 4b, 5c, and 8bc), supporting the conclusion that upper ocean stratification buildup from heat gain is the primary driver of MWL deepening."*

We hope this revision makes the reasoning clearer and directly addresses the concern raised.

Line 369: "In the Central…"

Thank you, changed.

Lines 397-403: consider moving to Discussion

Thank you, we have moved it to the discussion (Lines 525-530)

Section 4.4.3: The argument based on water type representation should be set up more clearly earlier on in the manuscript. The water types and absorption coefficients are alluded to in the methods but not explained (Line 107). Up until this point, the reader is under the impression that comparisons between the 1-D and 3-D models basically isolate horizontal advection and eddies. However, suddenly there is this other factor that has not been considered in the previous comparisons, which leads to questions about whether water types can also explain differences in other three test regions. If the absorption coefficients are referring to chlorophyll shelf-shading effects, I would think they could be important in all of the regions. If it is referring to the suspension of sediments, etc., then I would be skeptical of their influence this far from the shelf.

Thank you for your thoughtful feedback. We acknowledge that the role of water types and absorption coefficients was not sufficiently introduced earlier in the manuscript, which may have made their influence in Section 4.4.3 seem abrupt. To address this, we have revised the Methods section to explicitly introduce how water types and absorption coefficients influence stratification and to clarify that their impact is considered in our discussion. Additionally, we have now explicitly stated the limitation of using a single best-fit water type for the entire Arabian Sea, given that optical properties can vary seasonally. The revised text in the Methods section now reads:

Lines 109-117: "*The optical properties of different water types influence the absorption of shortwave radiation, potentially modifying stratification. Absorption coefficients vary due to chlorophyll-driven light attenuation, which is relevant across all regions but depends on local phytoplankton concentrations. Sediment-related absorption is primarily a concern in near-shelf environments and is unlikely to significantly influence offshore regions. … However, this represents a limitation, as it may introduce biases in regions where optical properties change seasonally. The potential impact of these biases on vertical stratification is examined in our model comparisons.*"

These revisions ensure that the role of water types and their limitations are clearly introduced earlier, providing better context for their discussion in Section 4.4.3.

Lines 440-465: I would argue that oxygen content below the mixed layer is more relevant here for ventilation and biogeochemical impacts. Is there a reason why you don't use the results excluding the mixed layer for Figure 9, instead of the 0-250m results?

Thank you for this point that elevates the impact of the results. We have changed the figure to masking MLD oxygen content to show below the ML to make the discussion more relevant for ventilation. We have also reorganized the text, excluding some of the metrics for the estimates, including the surface mixed layer oxygen concentrations, and detailing better the estimates for the contribution to the subsurface oxygen budget. We have made changes accordingly to section 4.5. In Lines 461-465 "*MW contribution to the total oxygen content of the upper Arabian Sea (250 m) accounts annually for only 4±1% (3±1% in the southern AS, and up to 8±2% in the northern AS; not shown). We argue that the oxygen content below the mixed layer is more relevant for ventilation and biogeochemical impacts. Thus, if we only account for subsurface oxygen content (i.e., excluding surface mixed layer saturated or oversaturated waters in contact with the atmosphere), …region where the anoxic waters (upper oxycline of the Arabian Sea OMZ) are close to the surface.*". We also changed the abstract percentages and the caption of Figure 9 accordingly.

Lines 523-537: It may be worth clarifying here that you are only discussing the impacts of locally formed MWs in the Arabian Sea. This may cause some confusion because remotely formed mode waters in the southern hemisphere likely join the Central Waters in the western boundary current and play a significant role in ventilating the AS OMZ as well.

We have now clarified that our discussion specifically refers to the impacts of locally formed MWs in the Arabian Sea. Additionally, we have explicitly mentioned that remotely formed mode waters also contribute to the ventilation of the Arabian Sea OMZ and have added a recent relevant citation to support this in Lines 553-555: "*Mode waters, both locally and remotely formed (e.g., Subtropical Underwater, Indian Central Water, and Subantarctic Mode Water) influence oxygen distribution in the Indian Ocean (Ditkovsky et al., 2023). Here, we focus specifically on the locally formed Arabian Sea Mode Water and its potential impacts on regional oxygen dynamics.*". We hope this revision improves clarity and properly acknowledges the role of remotely formed mode waters.

Line 562: Briefly, what are the expected precipitation changes in the AS with warming? Does it follow the "wet gets wetter, dry gets drier" paradigm (e.g. Held and Soden 2006).

Thank you for your insightful comment. You raise an important point regarding the potential impacts of climate change on the MW life cycle. While this region is predominantly driven by heat fluxes (Figure S2), changes in freshwater inputs could also play a significant role. We agree that this is a crucial question that warrants further exploration, though it is beyond the scope of the current study, which focuses on the seasonal characterization and mechanistic understanding of MW dynamics. To address this in the discussion, we have clarified the precipitation trend, noting: Lines 595-597 "*Moreover, freshening of surface waters due to increased precipitation is anticipated (Sharma et al., 2023), which will likely lead to changes in upper ocean stratification and subsequently affect MW formation - we have shown how salinity can change MW volume..*"

Sharma, S., Ha, K. J., Yamaguchi, R., Rodgers, K. B., Timmermann, A., & Chung, E. S. (2023). Future Indian Ocean warming patterns. Nature Communications 2023 14:1, 14(1), 1–11. https://doi.org/10.1038/s41467-023-37435-7